# Axonal G3BP1 stress granule protein limits axonal mRNA translation and nerve regeneration

Pabitra K. Sahoo [1], Seung Joon Lee[1], Poonam B. Jaiswal[2], Stefanie Alber[3], Amar N. Kar[1], Sharmina Miller-Randolph[1], Elizabeth E. Taylor[1], Terika Smith[1], Bhagat Singh[4], Tammy Szu-Yu Ho[4], Anatoly Urisman [5], Shreya Chand[5], Edsel A. Pena[6], Alma L. Burlingame[5], Clifford J. Woolf [4], Mike Fainzilber [3], Arthur W. English[2] & Jeffery L. Twiss [1]

Critical functions of intra-axonally synthesized proteins are thought to depend on regulated recruitment of mRNA from storage depots in axons. Here we show that axotomy of mammalian neurons induces translation of stored axonal mRNAs via regulation of the stress granule protein G3BP1, to support regeneration of peripheral nerves. G3BP1 aggregates within peripheral nerve axons in stress granule-like structures that decrease during regeneration, with a commensurate increase in phosphorylated G3BP1. Colocalization of G3BP1 with axonal mRNAs is also correlated with the growth state of the neuron. Disrupting G3BP functions by overexpressing a dominant-negative protein activates intra-axonal mRNA translation, increases axon growth in cultured neurons, disassembles axonal stress granule-like structures, and accelerates rat nerve regeneration in vivo.

[1] Department of Biological Sciences, University of South Carolina, Columbia 29208 SC, USA. [2] Department of Cell Biology, Emory University College of Medicine, Atlanta 30322 GA, USA. [3] Department of Biomolecular Sciences, Weizmann Institute of Science, Rehovot 76100, Israel. [4] FM Kirby Neurobiology Center and Boston Children's Hospital and Harvard Medical School, Boston 02115 MA, USA. [5] Department of Pharmaceutical Chemistry, University of California San Francisco, San Francisco 94158 CA, USA. [6] Department of Statistics, University of South Carolina, Columbia 29208 SC, USA. Correspondence and requests for materials should be addressed to J.L.T. (email: twiss@mailbox.sc.edu)

njured axons in the peripheral nervous system (PNS) use locally translated proteins for retrograde injury-signaling and regenerative growth[1]. Translation of axonal mRNAs can be activated by different stimuli including axotomy in mature neurons and in response to guidance cues in developing neurons[1], indicating that a significant fraction of axonal mRNAs are stored until a particular stimulus activates their translation. Stress granules (SG) serve as storage depots for mRNAs in non-neuronal systems, providing a mechanism to respond to cellular stress by sequestering unneeded mRNAs from translation[2]. Aggregation-prone mutations of the SG protein TIA1 and the RNA-binding protein TDP-43 have been shown to cause SG aggregation in neurons[3,4], but it is not known if SGs have roles in the normal function of neurons. Further, although SGs have been detected in dendrites[5], it is not clear if functional SGs are assembled in axons. The *RasGAP SH3 domain binding protein 1* (G3BP1) interacts with the 48S pre-initiation complex when translation is stalled[6], and it assembles SGs by virtue of its NTF2-like domain[7]. Murine G3BP1 knockout is embryonic lethal in 129/Sv mouse strain with CNS apoptosis[8], but not a mixed Balb/c/129/Sv background where altered synaptic plasticity and neuronal calcium homeostasis were seen[9]. This emphasizes roles for G3BP1 protein in the nervous system. Proteomics analyses recently reported G3BP1 interactomes from neurites of cultured motor neurons, where a core of SG-associated proteins was detected in the absence of stress[10]. Thus, G3BP1 aggregates may have functions in axons.

Here we show that translation of specific axonal mRNAs is negatively regulated in intact axons by G3BP1, and that this negative regulation is removed by dispersion of aggregated G3BP1 in regenerating peripheral nerves post injury to support accelerated axon growth. When phosphorylated on serine 149 (G3BP1$^{PS149}$), G3BP1's oligomerization is blocked and SGs disassemble, presumably releasing bound mRNAs for translation[7]. Loss of G3BP1 aggregation in SG-like structures in regenerating axons is accompanied by an increase in phosphorylated G3BP1. Disrupting G3BP1 function with a dominant-negative approach activates intra-axonal mRNA translation, increases axon growth in cultured neurons and accelerates nerve regeneration in vivo, and therefore represents a new pro-regenerative therapeutic approach.

## Results

### Axonal G3BP1 aggregates decrease during nerve regeneration.

We initially asked if axons of cultured primary sensory neurons contain stress granule-associated protein G3BP1. Sensory neurons in dissociated cultures from adult rat dorsal root ganglia (DRG) show strong immunoreactivity for G3BP1 in cell bodies and focally along their axons (Fig. 1a). By confocal microscopy, axonal G3BP1 signals appeared to show higher colocalization with other SG components compared with components of processing bodies (PB) that are linked to RNA degradation (Fig. 1b). Comparing Pearson's coefficients from these colabelings showed a significantly higher colocalization of axonal G3BP1 with SG markers than with PB markers (Fig. 1c). Colocalization of axonal G3BP1 with the SG protein HuR but not the PB protein DCP1a was further confirmed by proximity ligation assay (PLA; Fig. 1d). Overall, the axonal G3BP1 aggregates appeared smaller than those described for SGs in non-neuronal cells (diameter ~ 0.2–0.8 µm vs. ≥ 1 µm for stress-induced aggregates)[11], so the axonal SG-like entities approximate the ~ 250 nm diameter described for SG core structure[12].

Confocal microscopy of sciatic nerve sections showed robust, granular G3BP1 signals that overlapped with neurofilament (NF) across optical planes of the Z stacks (Fig. 1e). Signals for the SG

protein TIA1 focally overlapped with G3BP1, including the granular intra-axonal G3BP1 signals, and the axonal signals for both G3BP1 and TIA1 appeared to decrease in 7 d post-crush sciatic nerve just proximal to the crush site (Fig. 1e). The imaging parameters for these analyses were selected to visualize only the granular G3BP1 signals; on highly over-exposing the granular G3BP1 signals, more diffuse signals were noted for G3BP1 along the axon (data not shown), suggesting that the granular signals represent aggregates of G3BP1 in axons. We quantified the granular signals for G3BP1 and TIA1 in axons proximal to crush site at intervals over 3 h to 7 d after axotomy. Both proteins showed a striking increase in the intra-axonal signals at 3 h post-crush which fell to below the levels of the naive axons by 5 d post-crush; notably, the fold change for G3BP1 and TIA1 near perfectly overlapped across this time course (Fig. 1f). Axons are actively regenerating at 7 d after nerve crush (see below), and granular G3BP1 signals were largely excluded from the thin axons at the regenerating front of the injured nerve (Supplementary Fig. 1a).

Immunoblotting of DRG neurons transfected with control vs. G3BP1 targeting siRNAs confirmed the specificity of the anti-G3BP1 antibody used here (Supplementary Fig. 1b). To gain a more quantitative assessment of G3BP1 protein levels in axons, we used targeted mass spectrometry (MS) of sciatic nerve axoplasm taken over 3–28 d post injury. The MS analyses further confirmed presence of G3BP1 in axons and showed modest, but highly variable, declines in G3BP1 levels after an injury (Supplementary Fig. 1d). Approximately 3 cm of nerve proximal to the crush site was used for axoplasm preparations in these MS studies. Immunoblotting axoplasm from shorter segments of injured sciatic nerve (0 to −1 cm and −1 to −2 cm proximal to the crush site) showed a clear reduction in G3BP1 signals in 7 d injured compared to naive sciatic nerves (Supplementary Fig. 1e). Taken together, these data indicate that axonal SG-like structures and G3BP1 protein levels change after axonal injury and subsequent regeneration of PNS nerves. Thus, we wondered if the decrease in axonal SG-like aggregation might be a feature of growing axons. So we asked if axonal SG-like structures show alterations in vitro in DRG neurons with different axon growth capacity. DRG neurons that are conditioned by an in vivo crush injury 7 d prior to culture show more rapid axonal outgrowth over 18–48 h in vitro compared to uninjured (naive) DRGs[13], and the rapidly growing axons of those injury-conditioned neurons showed a decrease in G3BP1 aggregates compared to those of naive DRG cultures (Fig. 1g, h). Together, these data raise the possibility that aggregation of axonal G3BP1 in PNS axons is associated with a lower axon growth activity.

### G3BP1 is phosphorylated in regenerating axons.

Phosphorylation of G3BP1 on Serine 149 has been shown to trigger disassembly of SGs[7]. To determine if phosphorylation alters aggregation of axonal G3BP1, we expressed non-phosphorylatable and phosphomimetic G3BP1 mutants (G3BP1$^{S149A}$-GFP and G3BP1$^{S149E}$-GFP, respectively) in cultured DRGs. Axonal G3BP1$^{S149A}$-GFP showed aggregated signals that overlapped with the SG-associated protein HuR, while axonal G3BP1$^{S149E}$-GFP appeared diffuse (Fig. 2a, b, Supplementary Fig. 2a). G3BP1$^{S149E}$-GFP also showed significantly higher mobility in axons than G3BP1$^{S149A}$-GFP, and G3BP1-GFP showed mobility intermediate between G3BP1$^{S149E}$-GFP and G3BP1$^{S149A}$-GFP (Fig. 2c, Supplementary Fig. 2a). This is consistent with G3BP1$^{S149A}$-GFP aggregating into SG-like structures in axons.

We next asked whether endogenous G3BP1 is phosphorylated in axons using phospho-specific G3BP1$^{PS149}$ antibodies.

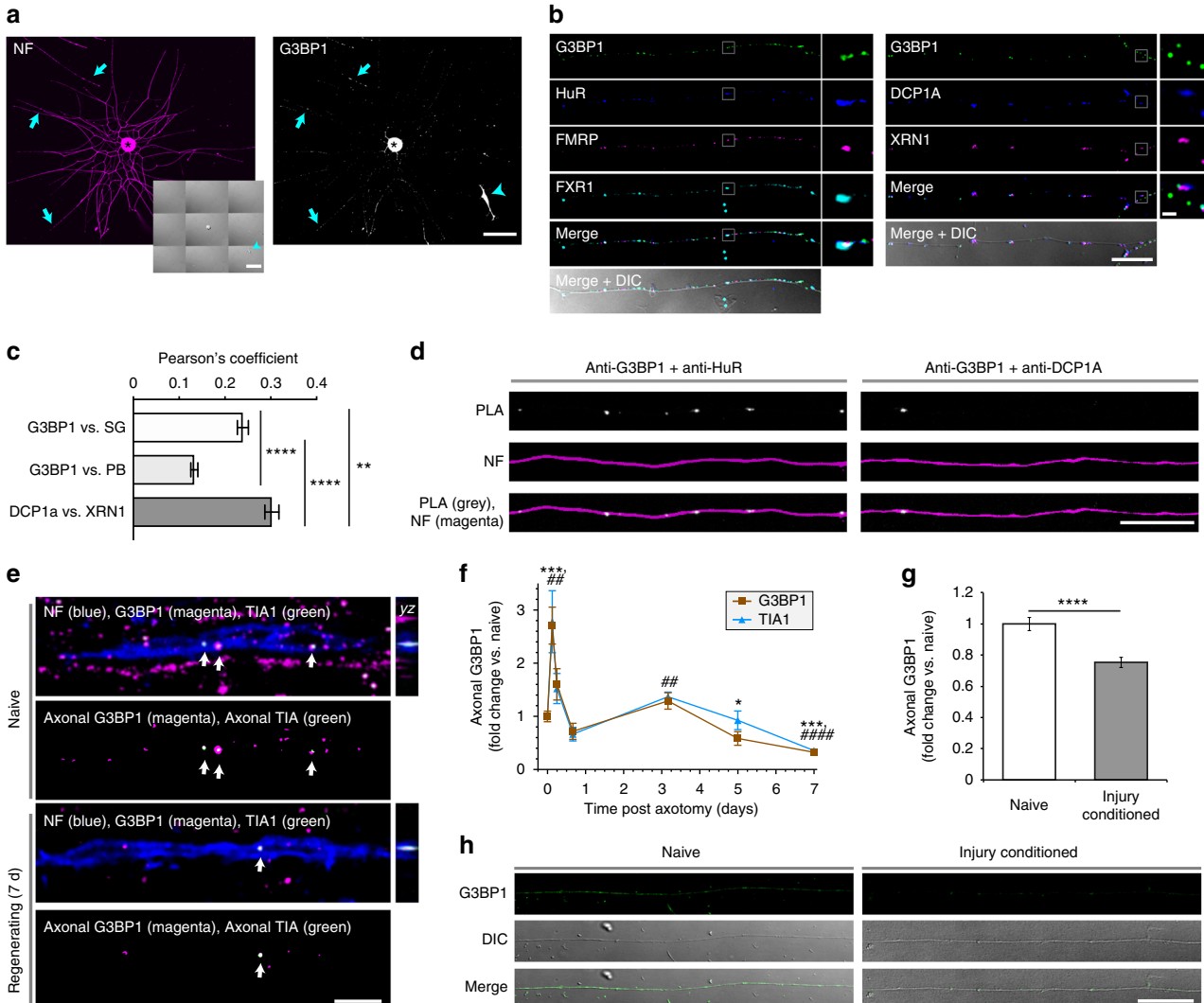

**Fig. 1** G3BP1 localizes to axons in stress granule-like aggregates. **a** Immunofluorescence for G3BP1 shows signals in the cell body (asterisk) and axons (arrows) of a cultured DRG neuron; arrowheads indicate Schwann cell with prominent G3BP1 immunoreactivity visible in the inset DIC image. Previous work has shown that neurites of these adult DRG neurons have axonal features and lack dendritic features[49]; we will use 'axon' for describing these hereafter [scale bar = 50 μm]. **b, c** Single planes for axons of naive DRG cultures co-labeled for indicated proteins are shown; box represents the area for high magnification insets to right (**b**). Axonal G3BP1 shows higher colocalization coefficients for SG than PB proteins by Fisher's Z transformation (**c**; $N \geq 30$ axons over 3 repetitions; $**p \leq 0.01$, $****p \leq 0.001$ by one-way ANOVA with Tukey HSD post-hoc) [scale bar = 10 μm for large panels, 1 μm for insets]. **d** PLA shows higher colocalization for G3BP1 and HuR than G3BP1 and DCP1A (G3BP1 + HuR PLA = 0.038 ± 0.003 and G3BP1 + DCP1A PLA = 0.027 ± 0.002 signals/μm$^2$; $N \geq 40$ neurons over 6 repetitions, $p = 0.016$ by Student's t-test) [scale bar = 20 μm]. **e, f** Confocal images for G3BP1 and TIA1 in naive and 7 d post-injured ('regenerating') sciatic nerve are shown (**e**). Upper image panels of each pair show G3BP1 and TIA1 merged with NF signals in single plane. Lower panels of each pair show XYZ for G3BP1 and TIA1 signals that overlap with NF across the Z stack Quantitation of axonal G3BP1 and TIA1 signals are shown (**f**) as mean ± SEM ($N = 6$ animals; $*p \leq 0.05$, $***p \leq 0.001$ for G3BP1 and $##p \leq 0.01$, $####p \leq 0.0001$ for TIA1 by Student's t-test for vs. naive) [scale bar = 5 μm]. **g, h** Quantification of G3BP1 levels (**g**) and G3BP1 immunofluorescence (**h**) in axons of DRGs cultured from naive vs. 7 d injury-conditioned animals are shown (mean ± SEM for $N \geq 66$ neurons over 3 repetitions; $***p \leq 0.001$ by Student's t-test) [scale bar 20 μm]

Immunoblotting with lysates from control vs. G3BP1 siRNA transfected DRGs showed a single band for anti-G3BP1$^{PS149}$ (Supplementary Fig. 1b). Treating DRG cultures with arsenic, a known inducer of SG aggregation[14], also decreased levels of G3BP1$^{PS149}$ without affecting overall G3BP1 levels by immuno-blotting (data not shown). By immunofluorescence, intra-axonal signals for anti-G3BP1$^{PS149}$ increased in proximal sciatic nerves 7 d post-crush injury (Fig. 2d, e). Thus, as the prevalence of axonal SG-like structures decreased in regenerating axons, there was a corresponding increase in axonal G3BP1$^{PS149}$. Moreover in cultured DRG neurons, the ratio of axonal G3BP1$^{PS149}$ to axonal G3BP1 aggregates increases in distal axons and growth cones

(Fig. 2f, g), suggesting that the axonal G3BP1 aggregation and phosphorylation are dynamically regulated along the growing axon.

**Axonal G3BP1 modulates axonal mRNA translation.** Previous studies detected ribosomes and translation factors in regenerating PNS axons in vivo[15], so the decrease in SG-like structures in distal axons could reflect increased protein synthesis in those axons. Thus, we asked if axonal mRNAs colocalize with G3BP1 in cultured neurons. Endogenous *Neuritin1* (*Nrn1*) and *Importin β1* (*Impβ1*) mRNAs showed clear colocalization with axonal G3BP1, but the mRNA encoding Growth-associated protein 43 (*Gap43*)

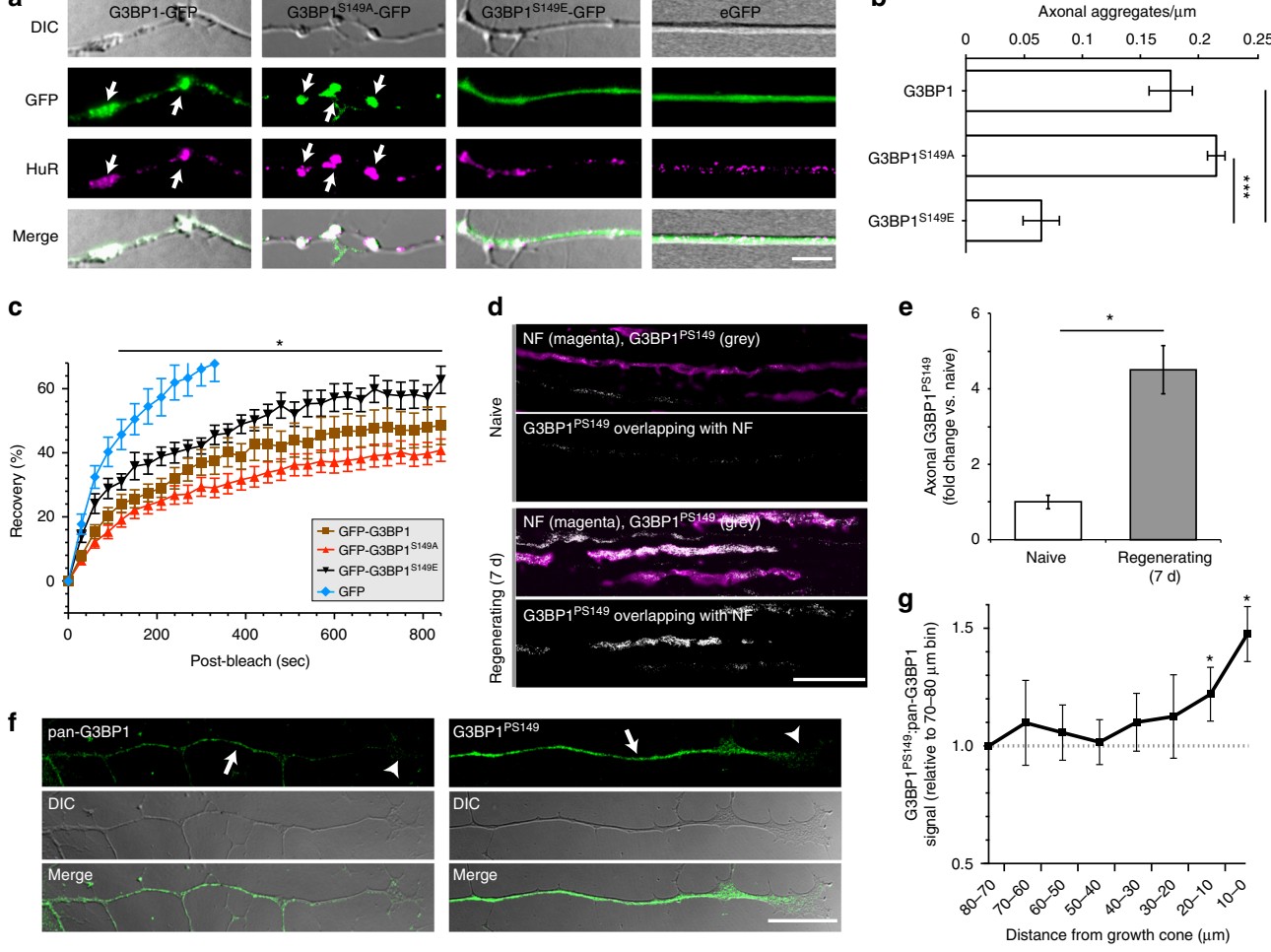

**Fig. 2** G3BP1 is phosphorylated in regenerating axons. **a** Representative images for axons of DRG neurons transfected with indicated G3BP1 constructs vs. eGFP are shown. G3BP1-GFP and G3BP1$^{S149A}$-GFP show prominent aggregates in axons that colocalize with HuR (arrows). In contrast, axonal signals for G3BP1$^{S149E}$-GFP and eGFP appear diffuse [scale bar = 5 µm]. **b** Quantification of axonal aggregates for G3BP1-GFP, G3BP1$^{S149A}$-GFP, and G3BP1$^{S149E}$-GFP is shown as average ± SEM ($N \geq 10$ neurons over 3 repetitions; ***$p \leq 0.005$ by one-way ANOVA with Tukey HSD post-hoc). **c** FRAP analyses for neurons transfected with constructs as in A are shown as average normalized % recovery ± SEM (see Supplementary Fig. 2a for representative FRAP image sequences). G3BP1$^{S149A}$-GFP shows much lower recovery than G3BP1$^{S149E}$-GFP; G3BP1-GFP is intermediate between G3BP1$^{S149A}$-GFP and G3BP1$^{S149E}$-GFP ($N \geq 13$ axons over 3 repetitions; *$p \leq 0.05$ between G3BP1$^{S149A}$-GFP vs. G3BP1$^{S149E}$-GFP by one-way ANOVA with Tukey HSD post-hoc). Only the 0–320 s. recovery signals for GFP are shown (at 840 s. GFP showed 85.5 ± 4.7% recovery with $p \leq 0.0001$ vs. G3BP1$^{S149E}$-GFP by one-way ANOVA with Tukey HSD post-hoc). **d–e** Exposure-matched confocal images for G3BP1$^{PS149}$ and NF are shown for sciatic nerve (**d**) as in Fig. 1e. There is a striking increase in G3BP1$^{PS149}$ immunoreactivity in the regenerating axons. Quantifications of these signals are shown as mean ± SEM (**e**; $N = 3$; *$p \leq 0.05$ by one-way ANOVA with Tukey HSD post-hoc) [scale bar = 20 µm]. **f–g** Distal axons of cultured DRGs immunostained with pan-G3BP1 vs. G3BP1$^{PS149}$ antibodies are shown as indicated (**f**). Aggregates of G3BP1 are visible in the axon shaft (arrow), but decrease moving distally towards the growth cone (arrowhead). G3BP1$^{PS149}$ signals are fairly consistent and extend into the growth cone (arrowhead). Quantification of signals (**g**) shows significant increase in ratio of G3BP1$^{PS149}$ immunoreactivity to G3BP1 aggregates moving distally to the growth cone ($N \geq 9$ neurons each over 3 repetitions; *$p \leq 0.05$ vs. 70–80 µm bin by one-way ANOVA with Tukey HSD post-hoc) [scale bar = 20 µm]

did not (Fig. 3a, Supplementary Fig. 3). The more rapidly growing axons of injury-conditioned DRG neurons showed higher colocalization of *Impβ1* with G3BP1 than those of naive DRGs, while axonal *Nrn1* showed the opposite (Fig. 3b). Axonal *Gap43* showed overall lower G3BP1 colocalization coefficients that did not change with injury conditioning (Fig. 3b). IMPβ1 protein is used for injury response after axotomy[16] and negatively regulates axon growth under basal conditions[17], while NRN1 protein supports regenerative growth of axons[18]. Thus, these distinct colocalizations of axonal *Impβ1* and *Nrn1* mRNAs with G3BP1 protein in naive vs. injury-conditioned neurons may reflect different functions of the encoded proteins in these different growth states.

We next used fluorescent reporters to determine if axonal SG-like structures contribute to translation. For this, we generated axonally targeted GFP$^{MYR}$ and mCherry$^{MYR}$ containing the 5′ and 3′ untranslated regions (UTR) of *Impβ1*, *Nrn1*, and *Gap43* mRNAs (GFP$^{MYR}$5′/3′impβ1, GFP$^{MYR}$5′/3′nrn1, and mCh$^{MYR}$5′/3′gap43, respectively; Fig. 3c). The membrane localizing myristoylation (MYR) of the fluorescent reporter proteins dramatically limits their diffusion from sites of translation, so GFP$^{MYR}$ and mCherry$^{MYR}$ proteins provide versatile reporters for localized protein synthesis in dendrites and axons using fluorescence recovery after photobleaching (FRAP)[19–22]. The 3′ (*Impβ1* and *Gap43*) and 5′ (*Nrn1*) UTRs provide axonal targeting for reporter mRNAs[16,18,23], and with both 5′ and 3′UTRs, the reporters approximate the translational regulation of the endogenous mRNAs. Recovery of axonal GFP$^{MYR}$5′/3′nrn1 and GFP$^{MYR}$5′/3′impβ1 fluorescence was decreased in DRGs

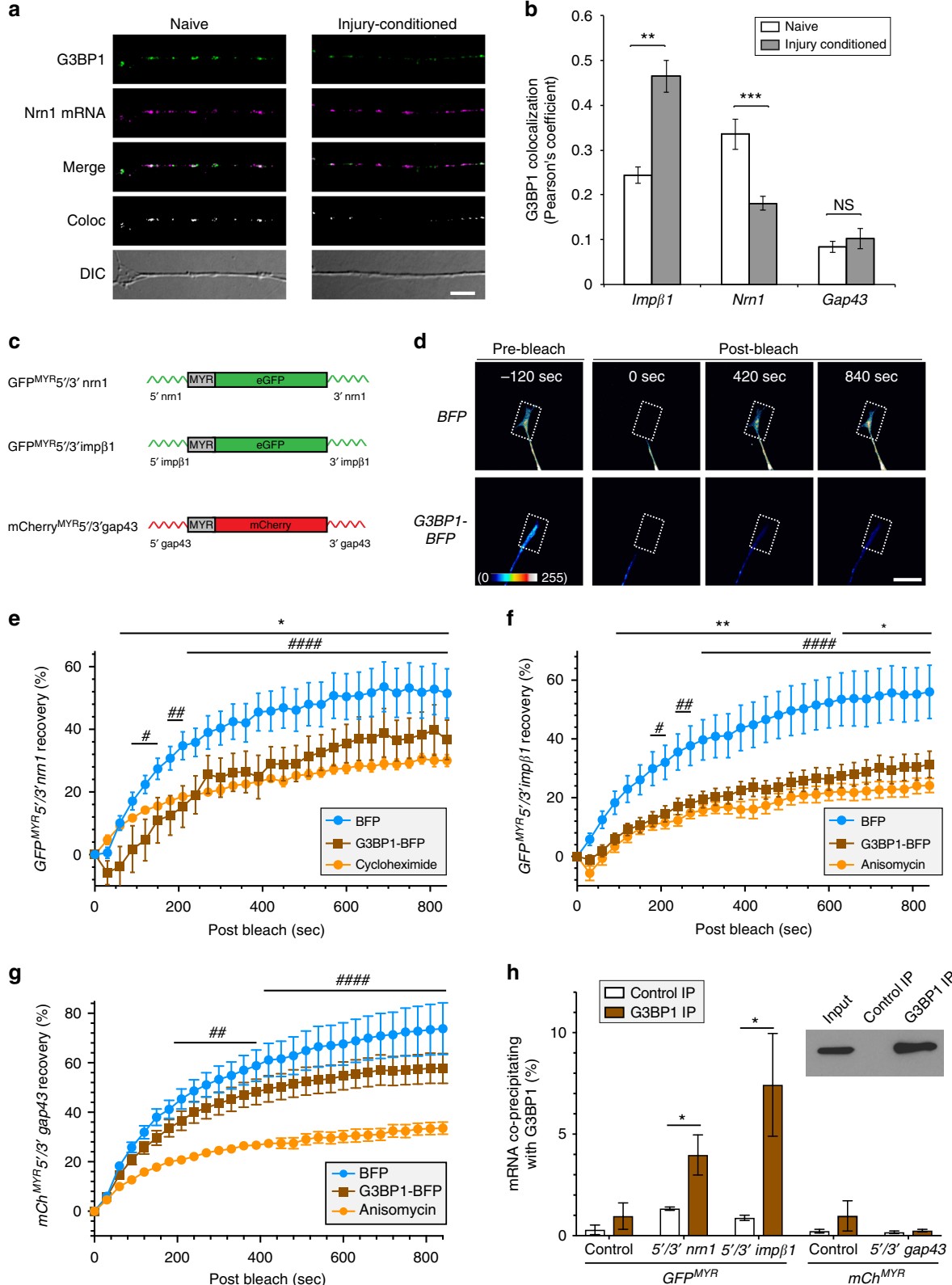

expressing G3BP1-BFP compared to the BFP control, but $mCh^{MYR}5'/3'gap43$ recovery was not significantly affected by G3BP1-BFP expression (Fig. 3d–g, Supplementary Movies 1–3). Treatment with translation inhibitors confirmed that the fluorescence recovery in axons after photobleaching represents new protein synthesis, and, interestingly, overexpression of

G3BP1-BFP approximated the effect of protein synthesis inhibition for $GFP^{MYR}5'/3'nrn1$ and $GFP^{MYR}5'/3'imp\beta1$ fluorescence recovery (Fig. 3e–g). Additionally, RNA immunoprecipitation (RIP) analyses showed enrichment of $GFP^{MYR}5'/3'imp\beta1$ and $GFP^{MYR}5'/3'nrn1$, but not $mCh^{MYR}5'/3'gap43$, in G3BP1 immunoprecipitates (Fig. 3h). Taken together, these data suggest that

**Fig. 3** G3BP1 regulates translation of axonal mRNAs. **a** Images of FISH/IF for *Nrn1* mRNA and G3BP1 protein are shown for axons of naive and 7 d injury-conditioned DRG neurons. Colocalization panel (Coloc) represents the mRNA:G3BP1 colocalization in a single optical plane (see Supplementary Fig. 3 for *Impβ1* and *Gap43* mRNA + G3BP1 colocalization images) [scale bar = 5 μm]. **b** Quantification of colocalizations for *Nrn1*, *Impβ1*, and *Gap43* mRNAs with G3BP1 in axons of neurons cultured from naive or 7 d injury-conditioned animals shown as average Pearson's coefficient ± SEM ($N \geq 21$ neurons over 3 repetitions; **$p \leq 0.01$ and ***$p \leq 0.005$ by one-way ANOVA with Tukey HSD post-hoc). **c** Schematics of translation reporter constructs used in panels **d**–**h** and Supplementary Movies 1–3. **d** Representative FRAP image sequences for DRG neurons co-transfected with GFP$^{MYR}$5'/3'nrn1 plus BFP or G3BP1-BFP. Boxed regions represent the photobleached ROIs. Videos for these images are included as Supplementary Movie 1 [scale bar = 20 μm]. **e**–**g** Quantifications of FRAP assays from DRGs expressing GFP$^{MYR}$5'/3'nrn1 (**e**) or GFP$^{MYR}$5'/3'impβ1 (**f**) or mCh$^{MYR}$5'/3'gap43 translation reporters along with G3BP1-BFP or control BFP are shown as normalized, average % recovery ± SEM ($N \geq 11$ neurons over 3 repetitions; *$p \leq 0.05$, and **$p \leq 0.01$ for BFP vs. G3BP1-BFP, #$p \leq 0.05$, ##$p \leq 0.01$, and ####$p \leq 0.0001$ for BFP vs. translation inhibitors by one-way ANOVA with Tukey HSD post-hoc). Representative videos for these FRAP sequences are included as Supplementary Movies 1–3. **h** HEK293T cells transfected with GFP$^{MYR}$5'/3'nrn1, GFP$^{MYR}$5'/3' impβ1, and mCh$^{MYR}$5'/3'gap43 show significant enrichment of GFP$^{MYR}$5'/3'nrn1 and GFP$^{MYR}$5'/3' impβ1 mRNAs coimmunoprecipitating with G3BP1 vs. control ($N = 4$ culture preparations; *$p \leq 0.05$ by Student's $t$-test). Western blot validating G3BP1 immunoprecipitation shown as inset. Values shown as average percent bound mRNA relative to input ± SEM

G3BP1 binds to *Nrn1* and *Impβ1* mRNAs and attenuates their translation in axons.

**The acidic domain of G3BP1 increases axonal growth.** Four domains have been defined for G3BP1 protein: an N-terminal NTF2-like 'A domain', a highly acidic 'B domain', a PxxP motif containing 'C domain', and a C-terminal RNA-binding motif containing 'D domain' (Fig. 4a)[7]. We acquired expression constructs for the B, C, and D domains and combinations of these to determine if they might affect the function of endogenous G3BP1 in the DRG neurons. Expression of these G3BP1 deletion constructs in naive DRG cultures showed that G3BP1 B, CD, BCD, and D domain proteins all localized to axons (Supplementary Fig. 4a). Neurons expressing the G3BP1 B domain showed significantly longer axons, while those expressing the D or CD domains showed shorter axons (Fig. 4b, Supplementary Fig. 4b). The G3BP1 D domain was previously shown to reduce protein synthesis in non-neuronal cells by triggering phosphorylation of the translation initiation factor eIF2α[24]. Interestingly, a combined construct of the B domain with the CD domain significantly increased axon outgrowth (Supplementary Fig. 4b), pointing to a dominant-negative effect of the B domain in absence of G3BP1's aggregating NTF-2 like region. Though the G3BP1 B domain contains Ser 149 whose phosphorylation causes SG disassembly[7], neither G3BP1$^{S149E}$-GFP nor G3BP1$^{S149A}$-GFP altered axon growth in the DRGs compared to GFP (Supplementary Fig. 4b). DRGs expressing the B domain- and CD domain-GFP showed modest decline in neurites per neuron, as did the expression of full-length G3BP1-GFP (Supplementary Fig. 4b). However, overexpression of full length G3BP1 had no significant effect on axon growth, perhaps indicating that G3BP1 is at saturating levels in DRG neurons. Consistent with this, siRNA-mediated G3BP1 depletion significantly increased axon growth and this was completely reversed by co-transfection with a siRNA-resistant G3BP1-GFP (Supplementary Fig. 4c, d). Co-transfecting with the G3BP1 B domain did not further increase axon length in the G3BP1 depleted neurons, suggesting that the B domain inhibits function of endogenous G3BP1.

In light of the axon growth-promoting effect of the G3BP1 B domain, we asked if introducing the G3BP1 B domain might alter axon regeneration in vivo. For this, adult rats were transduced with adeno-associated virus (AAV) expressing B domain, D domain, or full length G3BP1 and then subjected to sciatic nerve crush 7 d later. At 7 d after crush injury (14 d post-transduction), G3BP1-BFP, G3BP1 B domain-BFP, and G3BP1 D domain-BFP were visible in the regenerating sciatic nerve axons (Supplementary Fig. 5a). The G3BP1 B domain-BFP transduced animals showed significantly increased axon regeneration compared to G3BP1-BFP, and G3BP1 D domain-BFP, and GFP transduced

animals (Fig. 4c, Supplementary Fig. 5a,b). To test for the possibility of accelerated regeneration, we measured compound muscle action potentials (CMAP) in lateral gastrocnemius (LG) and tibialis anterior (TA) muscles to assess functional reinnervation after axotomy in control vs. B domain-transduced animals. Significantly accelerated recovery of CMAPs was seen with G3BP1 B domain expression in the LG at 4 and 6 wks and the TA at 4 wk after sciatic nerve crush, with control catching up by 8 wk in LG and 6 wk in TA (Fig. 4d, Supplementary Fig. 5c). The apparent faster recovery in the TA likely relates to the shorter regeneration distance and smaller muscle mass compared to the LG. Taken together, these data indicate that expression of the G3BP1 B domain accelerates peripheral nerve regeneration.

To determine if a smaller region of the G3BP1 B domain is sufficient to increase axon growth, we generated fluorescently-labeled, cell-permeable Tat fusion peptides corresponding to residues 147–166, 168–189, and 190–208 of rat G3BP1. These peptides each penetrated the neurons in DRG cultures by 30 min. after application (Supplementary Fig. 6a). When added to DRG cultures immediately after plating, both the 147–166 and 190–208 peptides increased axon length; the 190–208 peptide also increased the number of neurites per neuron (Supplementary Fig. 6b). Since the 190–208 peptide showed the longest axons and increased the overall number of neurites extended from each neuron, we focused our efforts on this peptide, in comparison to the 168–189 peptide that lacked activity. To discriminate between increased axon extensions vs. earlier initiation of axon growth, we exposed DRG cultures to peptides after the neurons had fully initiated axonal growth. With delayed application, the 190–208 peptide significantly increased axon length in both naive and pre-injured DRG neurons (Fig. 4e). E18 cortical neuron cultures also showed a significant increase in axon growth when the 190–208 peptide was applied to the axonal compartment of microfluidic culture devices (Fig. 4e, Supplementary Fig. 6c). Finally, the 190–208 peptide significantly increased neurite length in cultures of motor neurons generated from human induced pluripotent stem cells (Supplementary Fig. 6d). These data indicate that introducing amino acids 190–208 of rat G3BP1 increases axon growth in rodent and human neurons, and likely does so through an axon intrinsic mechanism(s).

**The G3BP1 acidic domain disassembles stress granule protein aggregates.** To determine if expression of the G3BP1 B domain interrupts the function of endogenous G3BP1, we asked if expressing the B domains alters axonal mRNA translation. Using a puromycinylation assay to test for translation of endogenous mRNAs[25], G3BP1 B domain expression led to significantly higher protein synthesis in axons but not cell bodies of cultured DRGs (Fig. 5a, b). Depletion of G3BP1 similarly increased protein

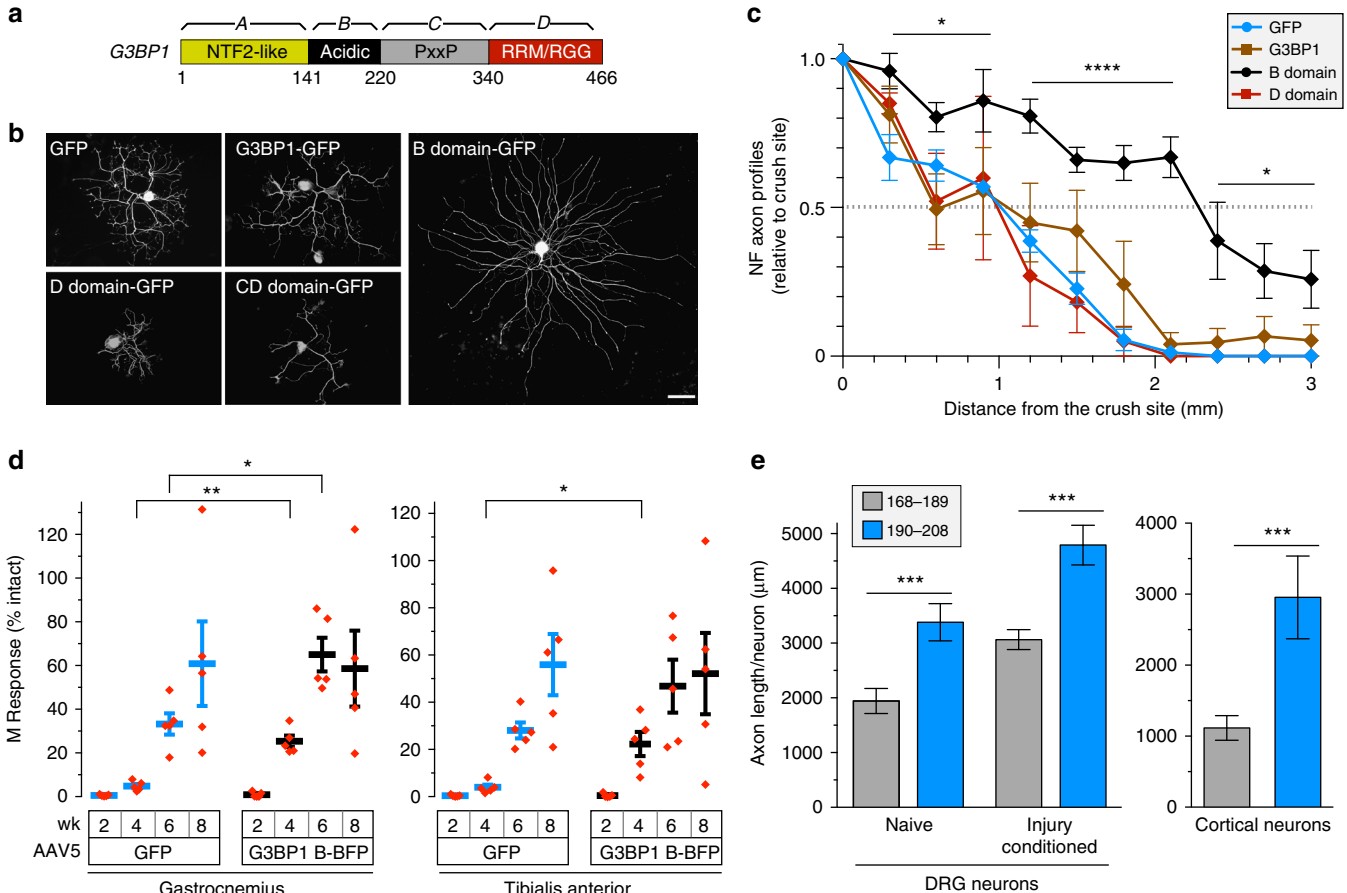

**Fig. 4** G3BP1 acidic domain expression accelerates nerve regeneration. **a** Schematic of G3BP1 domains as defined by Tourriere et al. (2003)[7]. **b** Representative images for NF-labeled DRG neurons transfected with indicated constructs are shown. Images were acquired at 60 h post-transfection [scale bar = 100 μm]. Axonal localization of these G3BP1-GFP domain proteins and quantitation of axon growth from G3BP1 domain-expressing DRG neurons are shown as Supplementary Fig. 4a-b. **c** Extent of axon regeneration at 7 d post sciatic nerve crush in adult rats transduced with AAV5 encoding G3BP1-BFP, G3BP1 B domain-BFP, G3BP1 D domain-BFP, or GFP control is shown as mean axonal profiles relative to crush site (0 mm) ± SEM. For representative images see Supplementary Fig. 5a ($N \geq 5$ animals per condition; * = $p \leq 0.05$ and **** = $p \leq 0.0001$ between B domain-BFP vs. GFP transduced animals by one-way ANOVA with Tukey HSD post-hoc). **d** Animals transduced with AAV5 encoding G3BP1 B domain-BFP vs. GFP were subjected to sciatic nerve crush and regeneration was assessed by muscle M response in tibialis anterior and gastrocnemius. Values are shown as average % intact M responses ± SEM (lines) with data points for individual animals plotted (*$p \leq 0.05$ and **$p \leq 0.01$ for B domain vs. GFP by Student's $t$-test for indicated data pairs). See Supplementary Fig. 5c for representative electrophysiological data. **e** Quantitation of axon growth from DRGs (left) and cortical neurons (right) treated with cell-permeable 168–189 or 190–208 G3BP1 peptides is shown. For DRGs, peptides were added to dissociated naive or 7 d injury-conditioned DRGs at 12 h and axon growth was assessed at 36 h in vitro. For cortical neurons, peptides were added to the axonal compartment of microfluidic devices at 3 d in vitro (DIV), and axon growth was assessed at 6 DIV. See Supplementary Fig. 6c for images of cortical cultures ($N \geq 95$ over 3 DRG cultures and 9 microfluidic devices over 3 cultures; ***$p \leq 0.005$ by one-way ANOVA with Tukey HSD post-hoc)

synthesis in the DRG axons with no significant effect on protein synthesis in the cell bodies (Fig. 5c, Supplementary Fig. 7a). Expression of the G3BP1 B domain also leads to increases in axonal but not in cell body levels of Nrn1 protein, without affecting axonal or cell body levels of Impβ1 or Gap43 proteins (Fig. 5d, Supplementary Fig 7b, c). Overexpression of full-length G3BP1 caused a decrease in axonal levels of both Nrn1 and Impβ1 proteins. Since both *Nrn1* and *Impβ1* mRNAs colocalized with G3BP1 and the *GFP*$^{MYR}$5′/3′*nrn1* and *GFP*$^{MYR}$5′/3′*impb1* reporter mRNAs coprecipitated with G3BP1, we asked if endogenous *Nrn1* and *Impβ1* mRNA binding to G3BP1 might be affected by the introduction of the B domain. Co-precipitation of these mRNAs with G3BP1-BFP was significantly reduced in neurons co-transfected with B domain-BFP construct (Fig. 5e). *GAP43* mRNA did show some binding to G3BP1-BFP, but this was not affected by the B domain expression, and none of these mRNAs precipitated with G3BP1 B domain-GFP or the control

GFP (Fig. 5e). These data suggest that the G3BP1 B domain increases axonal protein synthesis by causing release of mRNAs from axonal G3BP1 aggregates.

In light of the changes in translation upon expression of the G3BP1 B domain, we asked if the growth-promoting 190–208 peptide might also affect axonal protein synthesis. Treatment with the cell-permeable G3BP1 190–208 peptide significantly increased puromycin incorporation in DRG axons compared to untreated and G3BP1 168–189 peptide-treated cultures (Fig. 6a, b). The 190–208 peptide also significantly increased axonal recovery of *GFP*$^{MYR}$5′/3′*nrn1* fluorescence after photobleaching under control conditions and reversed the decrease in axonal translation seen with G3BP1 overexpression (Fig. 6c). The lack of effect previously observed for B domain expression on endogenous Impβ1 protein levels in axons was mirrored by the 190–208 peptide effects on *GFP*$^{MYR}$5′/3′*impβ1* and *mCh*$^{MYR}$5′/3′*gap43* translation in axons under control conditions. Moreover,

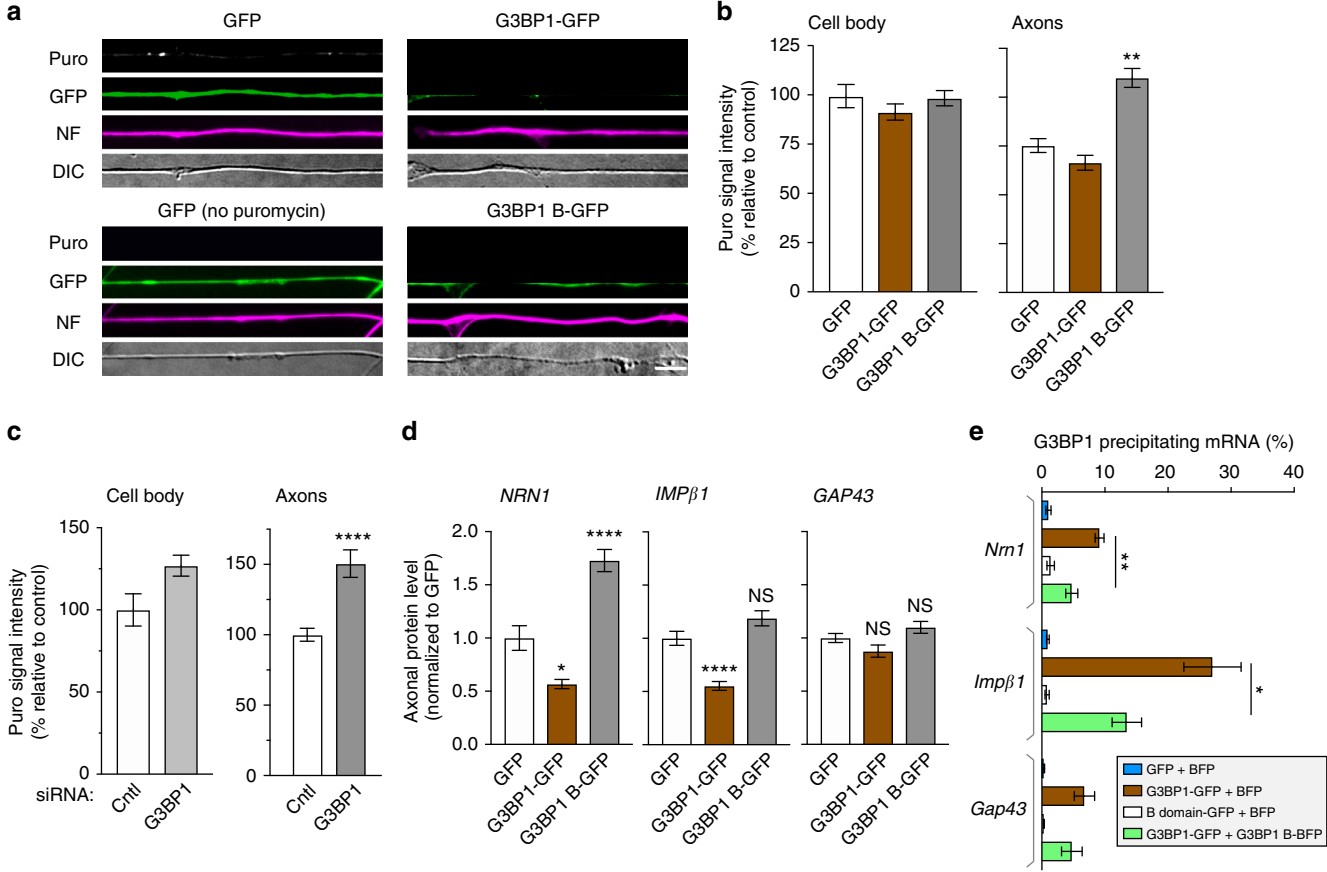

**Fig. 5** G3BP1 acidic domain increases axonal mRNA translation and disassembles stress granules. **a**, **b** Representative images for puromycin (Puro) incorporation in DRG neurons transfected with the indicated constructs are shown (**a**). Significant increase in axonal puromycin signals in the G3BP1 B domain-expressing neurons is seen, with no significant change in the cell body puromycin incorporation (**b**; $N \geq 23$ axons over three repetitions; **$p \leq 0.01$; ****$p \leq 0.0001$ by one-way ANOVA with Tukey HSD posthoc) [scale bar = 5 μm]. **c** G3BP1 depleted DRG cultures similarly show increased puromycin incorporation in axons with no significant change in cell body puromycin incorporation ($N \geq 23$ axons over three repetitions; **$p \leq 0.01$, ****$p \leq 0.0001$ by one-way ANOVA with Tukey HSD posthoc). See Supplementary Fig. 7a for representative images of these puromycin incorporation studies. **d** Quantitation of endogenous axonal NRN1, IMPβ1, and GAP43 protein levels in DRG cultures transfected with GFP, G3BP1-GFP, and G3BP1 B domain-GFP is shown. Axonal NRN1 and IMPβ1 but not GAP43 levels are significantly reduced in G3BP1 overexpression. G3BP1 B domain-expressing neurons show significantly higher axonal NRN1, but no change in axonal IMPβ1 and GAP43 levels ($N \geq 33$ axons over three repetitions; *$p \leq 0.05$, ****$p \leq 0.0001$ by one-way ANOVA with Tukey HSD posthoc). Representative images for axonal immunofluorescence and cell body NRN1, IMPβ1, and GAP43 proteins are shown in Supplementary Fig. 7b, c. **e** RTddPCR for axonal mRNAs co-precipitating with G3BP1-GFP in DRG neurons are shown as average % mRNA associated with G3BP1-GFP ± SEM. Nrn1 and Impβ1 mRNAs association with G3BP1-GFP significantly reduced by cotransfection with the G3BP1 B domain, but neither RNA coprecipitates with the B domain ($N = 4$ culture preparations; * $p \leq 0.05$, ** $p \leq 0.01$ by Student's t-test for the indicated data pairs)

the 190–208 peptide did not rescue the decline in axonal translation of $GFP^{MYR}5'/3'imp\beta1$ seen with G3BP1 overexpression (Fig. 6c). Together, these data indicate that disrupting G3BP1 function with overexpression of the B domain or the 190–208 G3BP1 peptide can specifically increase intra-axonal translation of some mRNAs.

Considering the effects of the G3BP1 B domain and 190–208 peptide on axonal mRNA translation, we reasoned that these agents might disrupt SGs. Indeed, G3BP1 B domain expression attenuated SG aggregation in NIH 3T3 cells exposed to sodium arsenite (Supplementary Fig. 8a, b). As noted above, arsenite is a potent inducer of SG aggregation[14], so we performed time-lapse imaging on DRG cultures to determine if the G3BP1 B domain affects the axonal SG-like structures. To this end, we treated DRG cultures with the cell-permeable 190–208 G3BP1 peptide, and monitored G3BP1-mCherry aggregates. The 190–208 peptide caused a striking decrease in axonal G3BP1-mCherry aggregates after 15 min. (Fig. 6d, e). Moreover, the remaining SG-like structures in axons were significantly smaller after 190–208

peptide treatment (Fig. 6f), and the remaining aggregates showed greater motility than in control conditions (Supplementary Movie 4). In contrast, the 168–189 G3BP1 peptide caused no change in SG density along the axons (Fig. 6d, e), and only modest but insignificant decrease in SG size (Fig. 6f). Together, these data indicate that the B domain expression and treatment with the 190–208 G3BP1 peptide disrupt aggregation of SG-like structures.

## Discussion

Many studies have now documented mRNA translation in axons, and this is particularly prominent in the PNS where intra-axonal protein synthesis contributes to axon regeneration after injury[1]. Some SG proteins have been detected in axons of PNS nerves, but intra-axonal functions for these proteins were only inferred from known functions of these proteins in other cellular systems[26]. Our data indicate that blocking G3BP1's function in the assembly of axonal SG-like structures increases intra-axonal protein synthesis

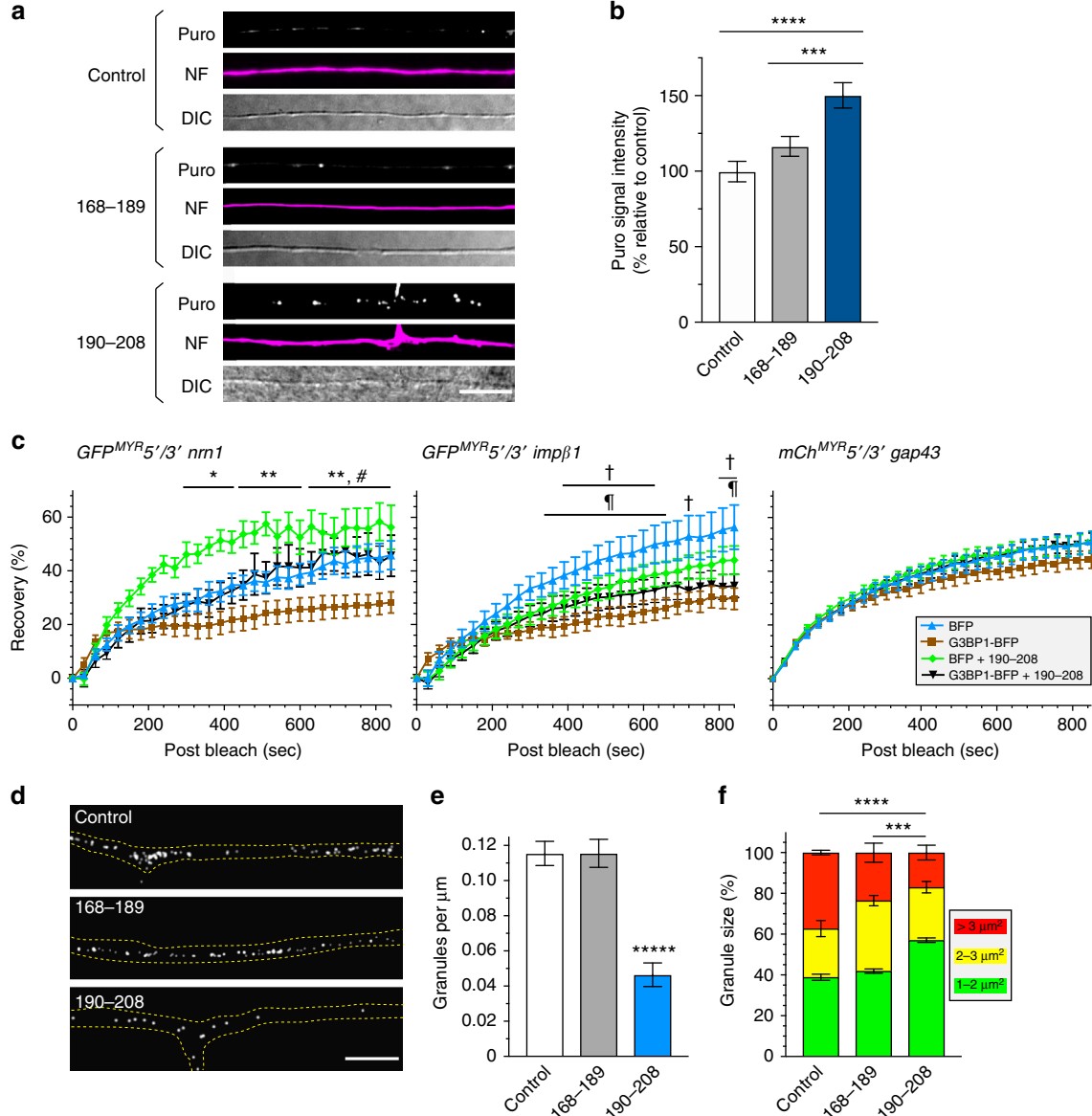

**Fig. 6** Cell permeable G3BP1 190–208 peptide increases axonal mRNA translation and disassembles stress granules. **a**, **b**, Representative images for puromycin incorporation in axons of control, 168–189 peptide and 190–208 peptide-treated DRG cultures are shown (**a**). Quantitation of puromycin incorporation into distal DRG axons under these conditions shows a significant increase in axonal protein synthesis for the 190–208 peptide-treated cultures compared to control and 168–189 peptide exposure (**b**; $N \geq 83$ axons over 3 DRG cultures; ***$p \leq 0.005$, ****$p \leq 0.0001$ by one-way ANOVA with Tukey HSD post-hoc). **c** FRAP analyses for DRGs for $GFP^{MYR}5'/3'nrn1$, $GFP^{MYR}5'/3'imp\beta1$ and $GFP^{MYR}5'/3'gap43$ in axons of DRGs expressing BFP or G3BP1-BFP $\pm 10$ μM 190–208 G3BP1 peptide (30 min. treatment). Only translation of $GFP^{MYR}5'/3'nrn1$ is increased by the 190–208 peptide with G3BP1 overexpression ($N \geq 11$ axons over three culture repetitions; all statistics were done by one-way ANOVA with Tukey HSD post-hoc: *$p \leq 0.05$, **$p \leq 0.01$ for BFP vs. BFP + 190–208 peptide; #$p \leq 0.05$ for G3BP1-BFP vs. G3BP1-BFP + 190–208; $p \leq 0.05$ for BFP vs. G3BP; and, †$p \leq 0.05$ for BFP vs. G3BP1-BFP + 190–208 peptide; no values for $GFP^{MYR}5'/3'gap43$ were statistically significant). **d** Representative images of G3BP1-mCh in DRG axons under control conditions and after treatment with 190–208 G3BP1 or 168–189 peptides for 15 min. are shown. Axon tracing was generated from DIC images [scale bar = 10 μm]. For representative live cell imaging sequence refer to Supplementary Movie 4. **e** Density of G3BP1-mCh aggregates along 100 μm length axons from DRG cultures treated as in **d** is shown ($N \geq 38$ axons over three repetitions; ****$p \leq 0.0001$ by ANOVA with Tukey HSD post-hoc). **f** Size of G3BP1-mCh aggregate is shown as indicated bins for from DRG cultures treated as in d ($N \geq 221$ aggregates over three repetitions; ****$p \leq 0.0005$, *****$p \leq 0.0001$ for entire population distributions by Kolmogorov–Smirnov test)

and accelerates PNS axon regeneration. Thus, axonal G3BP1 is a negative modulator of intra-axonal protein synthesis and axon growth. Since several thousand mRNAs have been identified in axons of cultured neurons[27,28], it is likely that translation of many axonal mRNAs could be regulated by G3BP1, as we show here for $Imp\beta1$ and $Nrn1$ mRNAs. The colocalization of different mRNAs

with these G3BP1 aggregates correlates with the growth status of neurons, and blocking G3BP1 aggregation provides a novel strategy to accelerate regeneration.

Assembly of SGs has been well characterized in response to different metabolic and oxidative stressors in non-neural systems[2,29–31]. The rapid increase in SG-like structures seen

here by 3 h after axotomy could reflect a stress response by the PNS axons, with a decrease in SG-like aggregates during axon growth at later time points. The decrease in G3BP1 aggregation was accompanied by an increase in phospho-G3BP1. Casein kinase 2 and AKT have recently been reported to phosphorylate G3BP1 on Ser 149 in other cellular systems[32,33]. Both of these proteins are present in axons, and it will be of interest for future work to determine their roles in intra-axonal signaling cascades regulating G3BP1 phosphorylation. Notably, we see aggregates of G3BP1 and TIA1 in uninjured PNS axons and G3BP1 aggregates in axon shafts of cultured neurons with growing axons. We suspect that these structures correspond to the 'core SGs' that have been defined in other cell types[12]. Recent proteomics analyses for SG protein interactomes, including the G3BP1 protein analyzed herein, point to pre-assembly of some SG proteins under non-stress conditions[10,34]. These interactomes also included HuR (also known as ELAVL1), FXR1, FMRP, and TIA1 proteins that we show colocalize with axonal G3BP1 protein. Consistent with the possibility of a core SG present in uninjured and growing axons, as our study suggests, core SG components were recently shown to interact in neurites of human IPSC-derived motor neurons before application of arsenic[10].

Our study notably shows that axonal G3BP1 and TIA1 do not always colocalize. Likewise, the Pearson's coefficient for colocalization of G3BP1 with SG components is low despite being statistically higher than the coefficients for G3BP1 with PB proteins. Overexpression of G3BP1 was shown to precipitate SG assembly in the absence of any stress in non-neuronal cells[7]. Only some of the aggregates seen with overexpressed G3BP1 colocalize with TIA1, but G3BP1-associating mRNAs were found in both TIA1-positive and TIA1-negative G3BP1 aggregates[35]. Thus, it is likely that the axonal G3BP1 aggregates seen here that are separate from TIA1 can also interact with mRNAs. Regardless of whether the axonal G3BP1 aggregates are classic SGs or even core SG aggregates, our data clearly show these axonal aggregates attenuate axonal protein synthesis and limit rates of axon growth, so the axonal G3BP1 aggregates are biologically significant. Future studies will be needed to compare and contrast the constituents of these axonal G3BP1 aggregates to those of classic SGs.

The translation of different axonal mRNAs will undoubtedly show a high degree of regulatory complexity, and it is likely that numerous axonal mRNAs could be regulated by G3BP1 interactions. Interestingly, all axonal mRNAs were not regulated by the G3BP1 aggregates, since Gap43 showed comparatively low interaction with G3BP1 and its translation was not affected by G3BP1 overexpression or B domain manipulations. This may reflect differences in post-transcriptional regulation between Gap43, Nrn1 and Impβ1 mRNAs in axons. Impβ1 mRNA is constitutively transported into axons[36], where its localized translation is activated through $Ca^{2+}$-dependent pathways after axotomy[22]. In contrast, Nrn1's transport into axons is increased after axotomy, with the mRNA shifting from soma-predominant to axon-predominant during regeneration[18]. On the other hand, Gap43's transcription is increased approximately 5 fold after axotomy, with increased axonal localization commensurate with an overall increase in Gap43 levels[23]. Based on the low colocalization of Gap43 with G3BP1 and the lack of effect of G3BP1 on its translation, Gap43 mRNA does not seem to be regulated by the axonal SG-like structures. It is intriguing to speculate that differences in transcriptional vs. post-transcriptional regulation contribute to whether individual mRNAs are regulated by the axonal SG-like structures. Such distinctions would segregate the axonal transcriptome into mRNA cohorts based on the mechanisms of their transcriptional and/or translational regulation and axonal transport.

The difference between Impβ1 and Nrn1 colocalization with G3BP1 in naive vs. injury-conditioned neurons likely reflects different needs for the corresponding proteins in different growth states. DRG neurons that are pre-injured by an in vivo axotomy days prior to culturing show rapid elongation of relatively unbranched axons that is transcription independent[13]. This rapid axonal growth occurs through translational control of existing mRNAs[37], and the injury-conditioned neurons show higher intra-axonal protein synthesis than naive DRG neurons[38]. Nrn1 protein promotes neurite growth, and increasing axonal targeting of Nrn1 mRNA increases axon growth[39]. Hence, the decrease in Nrn1 mRNA associated with SG-like aggregates in axons of injury-conditioned neurons would free the mRNA for translation to promote axon growth. On the other hand, Impβ1 mRNA translation is induced by axotomy, with its protein product providing a retrograde signal to activate regeneration-associated gene expression in the soma[16]. Continued translation of Impβ1 mRNA likely decreases axon elongation due to its role in axon length sensing[17]. Consequently, rapid axon growth after injury conditioning could also be facilitated by sequestering Impβ1 mRNA from translation.

Both Nrn1 and Impβ1 mRNAs were released from interaction with G3BP1 when the G3BP1 B domain was introduced. Since the B domain did not co-precipitate these mRNAs, the release of mRNAs from G3BP1 interaction is not via a competitive interaction for mRNA binding by the B domain with full length G3BP1. Though axonal translation of both Nrn1 and Impβ1 was decreased by overexpression of G3BP1, only Nrn1 showed increased translation in response to B domain expression and 190–208 peptide treatment. This indicates that the release of stored mRNAs in axons is not sufficient for their translation. Additional stimuli are undoubtedly needed to translate Impβ1 mRNA compared to Nrn1 mRNA. Impβ1 mRNA was initially shown to be translated in axons after injury and this requires an increase in axoplasmic $Ca^{2+}$ levels[40]. Increased $Ca^{2+}$ is known to trigger phosphorylation of eIF2α[41], and phosphomimetic eIF2α was shown to increase the translation of axonal Calr and Hspa5 mRNAs in cultured DRG neurons[42]. Thus, axoplasmic $Ca^{2+}$ levels may provide one regulatory mechanism for determining which mRNAs are translated upon release from the axonal SG-like structures. Differential susceptibility to mTOR regulation may provide an additional layer of regulation, as frequently reported for survival promoting retrograde injury signals in peripheral nerve[43].

In summary, our study points to axonal G3BP1 as a specific modulator of intra-axonal protein synthesis and axon growth. Since G3BP1 is aggregated in uninjured PNS axons, our data point to unrealized functions for SG-like aggregates in axons under non-stress conditions. Preventing this SG-like aggregation of axonal proteins during regeneration increases the rate of axon regrowth. Considering that Tat fusion peptides for NR2B9c have been used in a clinical trial for ischemic protection during endovascular repair for intracranial aneurysms[44], the growth-promoting effects of the cell-permeable 190–208 G3BP1 peptide may represent a novel therapeutic lead for accelerating nerve regeneration. Since peripheral nerves typically regenerate at only 1–2 mm per day, accelerating axon growth rates by interfering with axonal G3BP1 function could significantly shorten recovery times and allow axons to reach a more receptive environment to reinnervate target tissues.

## Methods

**Animal use and survival surgery.** Institutional Animal Care and Use Committees of University of South Carolina, Emory University, and Weizmann Institute of Science approved all animal procedures. Male Sprague Dawley rats (175–250 g) were used for all sciatic nerve injury and DRG culture experiments. Embryonic day

18 (E18; male and female) rat pups were used for cortical neuron culture experiments. Isofluorane was used for anesthesia for AAV transduction and peripheral nerve injuries, and ketamine plus xylazine was used for electrophysiology studies (see below).

For peripheral nerve injury, anesthetized rats were subjected to a sciatic nerve crush at mid-thigh as described[37]. In cases where animals were transduced with virus prior to injury, AAV5 was injected into the proximal sciatic nerve 7 d prior to crush injury (at sciatic notch level; $9–14 \times 10^{10}$ particles in 0.6 M NaCl).

Axoplasm was obtained from sciatic nerve at 3–28 d after crush injury at mid-thigh level. Approximately 3 cm segments of nerve proximal to the injury site (or equivalent level on contralateral [naive] side) were dissected and axoplasm extruded into 20 mM HEPES [pH 7.3], 110 mM potassium acetate, and 5 mM magnesium acetate (nuclear transport buffer) supplemented with protease/phosphatase inhibitor cocktail (Roche) and RNasin Plus (Promega)[40]. After clearing by centrifugation at 20,000 ×*g*, 4 °C for 30 min., supernatants were mixed with 3 volumes of Trizol LS (Invitrogen) and processed for mass spectrometry (see below). 3 animals were used for each time point.

**Cell culture**. For primary neuronal cultures, L4–5 DRG were harvested in Hybernate-A medium (BrainBits) and then dissociated as described[37]. After centrifugation and washing in DMEM/F12 (Life Technologies), cells were resuspended in DMEM/F12, 1 x N1 supplement (Sigma), 10% fetal bovine serum (Hyclone), and 10 μM cytosine arabinoside (Sigma). Dissociated DRGs were plated immediately on laminin/poly-L-lysine-coated coverslips or transfected (see below) and then plated on coated coverslips.

For cortical neuron cultures, E18 cortices were dissected in Hibernate E (BrainBits) and dissociated using the Neural Tissue Dissociation kit (Miltenyi Biotec). For this, minced cortices were incubated in a pre-warmed enzyme mix at 37 °C for 15 min; tissues were then triturated and applied to a 40 μm cell strainer. After washing and centrifugation, neurons were seeded at a density of $1 \times 10^5$ cells per poly-D-lysine-coated microfluidic device (Xona Microfluidics). NbActive-1 medium (BrainBits) supplemented with 100 U/ml of Penicillin-Streptomycin (Life Technologies), 2 mM L-glutamine (Life Technologies), and 1 X N21 supplement (R&D Systems) was used as culture medium.

Human induced pluripotent stem cells (hiPSCs) were maintained in dishes coated with Matrigel (Corning) in Flex8 media (ThermoFisher). hiPSCs were differentiated into human motor neurons using a directed differentiation protocol optimized by Kevin Eggan[45]. 7000 neurons/well were plated on laminin- or CSPG-coated 96 well plates. 100 μl at 10 μg/ml Laminin (ThermoFisher) and 25 ng CSPGs (Millipore) were used per well.

NIH-3T3 and HEK293T cells were maintained in DMEM (Life Technologies) supplemented with 10% FBS (Gibco) and 100 U/ml of Penicillin-Streptomycin (Life Technologies).

For DRG neuron transfection, dissociated ganglia were pelleted by centrifugation at 100 ×*g* for 5 min and resuspended in 'nucleofector solution' (Rat Neuron Nucleofector kit; Lonza). 5–7 μg plasmid was electoporated using an AMAXA Nucleofector apparatus (program SCN-8; Lonza). For siRNA transfection, 100 nM siRNAs (Dharmacon) were used with DharmaFECT 3 reagent and incubated for 36 h. A 3′UTR targeted siRNA (5′ CCACAUAGGAGCUGGGAAUUU 3′) was used for depleting G3BP1 for experiments assessing axon growth where siRNA-resistant G3BP1 constructs were used for rescue. Dharmacon On-target plus-SMART pool siRNA (Cat no. L-101659-02-0005) against *G3bp1* was used in antibody specificity testing and Puromycinylation assays. Non-targeting siRNAs were as control. RTddPCR and immunoblotting was used to test the efficiency of G3BP1 depletion (see below). HEK293T cells were transfected using Lipofectamine® 2000 per manufacturer's instructions (Invitrogen).

AAV5 preparations were titrated in DRG cultures by incubating with $1.8–2.8 \times 10^{10}$ particles of AAV5 overnight.

For arsenic treatment to induce SG aggregation, transfected NIH3T3 cells were grown to 60–80% confluence and were then treated with 0.5 mM sodium arsenite (Sigma) for 30 min.

For peptide treatments, 10 μM Tat-fused peptides were added to dissociated DRG cultures at 2 or 12 h after plating. Neurite outgrowth was assessed 24 h after addition of peptides. For the cortical cultures, 10 μM peptide was applied to the axonal compartment at 3 d in vitro (DIV) and axonal growth was assessed at 6 DIV. For iMotor neurons, 20 μM peptides were immediately added and neurite growth was assessed 24 h later.

**Plasmid and viral expression constructs**. All fluorescent reporter constructs for analyses of RNA translation were based on eGFP with myristoylation element (GFP[MYR]; originally provided by Dr. Erin Schuman, Max-Plank Inst., Frankfurt)[20] or mCherry plasmid with myristoylation element (mCh[MYR])[42]. Reporter constructs containing 5′ and 3′UTRs of rat *Nrn1* and *Gap43* mRNAs have been published[18,23]. For *Impβ1*, the rat 5′UTR was cloned by PCR and inserted directly upstream of the initiation codon in GFP[MYR]3′impβ1[17].

Human G3BP1 wild type, S149A, S149E and deletion constructs as GFP-tagged proteins were generously provided by Dr. Jamal Tazi, Institut de Génétique Moléculaire de Montpellier[7]. The G3BP1-mCherry construct was generated by PCR, amplifying G3BP1 coding sequence with 5′ *NheI* and 3′ *HindIII* restriction

sites. After *NheI* + *HindIII* digestion, G3BP1 CDS was subcloned into *NheI* + *HindIII*-digested pmCherry-N1 vector (Clontech).

AAV5 preparations were generated in UNC Chapel Hill Viral Vector Core. All plasmid inserts were fully sequenced prior to generating AAV. *BglII* + *XhoI* digested human G3BP1 cDNA (from pGFP-G3BP1) was subcloned into *BamHI* + *XhoI* digested pAAV-cDNA6-V5His vector (Vector Biolabs). G3BP1 deletion constructs were amplified by PCR with terminal *HindIII* and *XhoI* restriction sites (primer sequences available on request). After digestion with *HindIII* and *XhoI*, products were cloned into *HindIII* + *XhoI*-digested pAAV-cDNA6-V5His vector. BFP was excised from the pTagBFP-N vector (Evrogen) using *EcoRI* + *NotI* and ligated in-frame directly 3′ to the G3BP1 sequences in pAAV-cDNA6-V5His.

**Generation of Tat-tagged G3BP1 B domain peptides**. Three peptides were generated from the rat G3BP1 B domain sequence (amino acids 140–220; UniProt ID # D3ZYS7_RAT) by Bachem Americas, Inc. Peptides were synthesized with N-terminal dansyl chloride or FITC and N- or C-terminal HIV Tat peptide for cell permeability[46]; the Tat sequence was placed at the least conserved end of the sequence based on P-BLAST of vertebrate G3BP1 sequences available in UniProt database. Peptide sequences were (Tat is shown in italics): 147–166, EESEEEVE EPEENQQSPEVV-*YGNKKNNQNNN*; 168–189, DDSGTFYDQTVSNDLEEH LEEP-*YGNKKNNQNNN*; and 190–208, *YGNKKNNQNNN*-VVEPEP EPEPEPEPEPVSE.

**Immunofluorescent staining**. All procedures were performed at room temperature (RT) unless specified otherwise. Cultured neurons were fixed in 4% paraformaldehyde (PFA) in phosphate-buffered saline (PBS) and processed as described[18]. Primary antibodies consisted of: rabbit anti-G3BP1 (1:200, Sigma), RT97 mouse anti-neurofilament (NF; 1:500, Devel. Studies Hybridoma Bank), goat anti-NRN1 (1:100, Novartis), rabbit anti-IMPβ1 (1:100, My Biosciences), rabbit anti-GAP43 (1:5000, Novus), and rabbit anti-G3BP1[PS149] (1:300, Sigma). FITC-conjugated donkey anti-rabbit and Cy3-conjugated donkey anti-mouse (both at 1:200, Jackson ImmunoRes.) were used as secondary antibodies.

For G3BP1 colocalization with SG and PB proteins, Zenon antibody labeling kit (Life Technologies) was used to directly label antibodies with fluorophores. Combinations of rabbit anti-G3BP1 (Sigma) + Alexa-488, rabbit anti-HuR (Millipore) + Alexa-405, rabbit anti-FMRP (Cell Signaling Tech) + Alexa-555, and rabbit anti-FXR1 (a kind gift from Dr. Khandiah, Institut Universitaire en Santé Mentale de Québec) + Alexa-633 or rabbit anti-G3BP1 + Alexa-488, rabbit anti-DCP1A (Abcam) + Alexa-405, and rabbit anti-XRN1 (Bethyl Lab) + Alexa-633 were used at 1:50 dilution for each antibody. Equal amounts of rabbit-IgG labeled with Alexa-405, -488, -555 and -633 were used as control.

For quantifying axonal content of G3BP1, TIA1, and G3BP1[PS149] in peripheral nerve, sciatic nerve segments were fixed for 4 h in 4% PFA and then cryoprotected overnight in 30 % sucrose, PBS at 4 °C. 10 μm cryostat sections were processed for immunostaining as previously described[18]. Primary antibodies consisted of rabbit anti-G3BP1 (1:100), rabbit anti-phospho-G3BP1[PS149] (1:100), and RT97 mouse anti-NF (1:300). Secondary antibodies were FITC-conjugated donkey anti-rabbit and Cy3-conjugated donkey anti-mouse (both at 1:200, Jackson ImmunoRes.). Immunoblotting confirmed the specificity of the anti-G3BP1 and -G3BP1[PS149] antibodies and by immunofluorescence signals for both antibodies were decreased in DRGs transfected with siRNAs to G3BP1 (Supplementary Fig 1b, c).

Paraffin sections were used for analyses of nerve regeneration. For this, 10 μm thick paraffin sections of sciatic nerve were deparaffinized in 100% xylene ($2 \times 10$ min) followed by 100% ethanol ($2 \times 10$ min). Sections were rehydrated by sequential incubations in 95, 75 and 50% ethanol for 5 min each, and then rinsed in deionized water. Sections were permeabilized in 0.3% Triton X-100 in PBS, and then rinsed in PBS for 20 min and equilibrated in 50 mM Tris [pH 7.4], 150 mM NaCl, 1% heat-shock bovine serum albumin (BSA), and 1% protease-free BSA (Roche) ('IF buffer'). Sections were then blocked in IF buffer plus 2% heat-shock BSA, and 2% fetal bovine serum for 1.5–2 h. After blocking, samples were incubated overnight at 4 °C in a humidified chamber with the primary antibodies in IF buffer. Samples were washed in IF buffer three times and then incubated with secondary antibodies diluted in IF buffer for 45 min. Samples were washed in IF buffer three times followed by a rinse in PBS and deionized water. Primary antibodies consisted of: RT97 mouse anti-NF (1:300) and rabbit anti-RFP (1:100, Rockland Immun. Chem.). The RFP antibody was confirmed to detect BFP by immunoblotting (see below) and immunolabeling of transfected DRG neurons (data not shown). Secondary antibodies were used as above.

All samples were mounted with Prolong Gold Antifade (Invitrogen) and analyzed by epifluorescent or confocal microscopy. Leica DMI6000 epifluorescent microscope with ORCA Flash ER CCD camera (Hamamatsu) or Leica SP8X confocal microscope with HyD detectors was used for imaging unless specified otherwise. For quantitation between samples, imaging parameters were matched for exposure, gain, offset and post-processing. For protein–protein colocalizations, HyVolution (Leica/Huygens) deconvolution was used to optimize optical resolution in confocal image stacks acquired with parameters optimized for this post-processing.

**Fluorescence in-situ hybridization (FISH)**. For FISH, DRG cultures were fixed for 15 min in 2% PFA in PBS. RNA-protein colocalization was performed using

custom 5′ Cy3-labeled 'Stellaris' probes (probe sequences available upon request; BioSearch Tech.)[17]. Scrambled probes were used as control for specificity; samples processed without the addition of primary antibody were used as control for antibody specificity. Primary antibodies consisted of rabbit anti-G3BP1 (1:100) and RT97 mouse anti-NF (1:200). FITC-conjugated donkey anti-rabbit and Cy5-conjugated donkey anti-mouse (both at 1:200) were used as secondaries. Samples were mounted as above and analyzed using a Leica SP8X confocal microscope. Samples were post-processed with HyVolution integrated into the Leica LAX software and analyzed as outlined below for RNA-protein colocalization.

**Proximity ligation assay (PLA)**. PLA has been used to show protein colocalization within a range of approximately 40 nm[17]. For this, we used Duolink kit per the manufacturer's instructions (Sigma). Briefly, dissociated DRGs were cultured for 48 h, fixed with 4% PFA in PBS. Samples were blocked and permeabilized in PBS plus 0.1% Triton X-100, 5% donkey serum, 1% BSA for 30 min. Samples were incubated with the following primary antibodies overnight at 4 °C in PBS plus 1% donkey serum: rabbit anti-G3BP (1:100), mouse anti-HuR (1:100), and mouse anti-DCP1a (1:100). After washing in PBS, samples were incubated with PLA reagent ± probes for 1 h at 37 °C. Following three washes in 0.01 M Tris [pH 7.4], 0.15 M NaCl and 0.05% Tween 20 ('buffer A'), ligation-ligase mix was applied and samples were incubated for 30 min at 37 °C. Subsequently, samples were washed 2 × in buffer A, then the amplification-polymerase mix was added and samples were incubated for 110 min at 37 °C. Finally coverslips were washed three times in 0.2 M Tris-Cl [pH 7.5], 0.1 M NaCl ('buffer B') and then incubated with chicken anti-NF H antibody (1:2000; Abcam) for 45 min at RT. Coverslips were washed in buffer B three times, incubated for 45 min with Alexa 488-conjugated donkey anti-chicken (Jackson ImmunoRes., 1:1000), washed and mounted with Mowiol. PLA with only one of the two primary antibodies (but adding both PLA probes) was used as a technical control.

Imaging was performed using an Olympus FV1000 confocal microscope (60 × / NA 1.35 UPLSAPO oil immersion objective). Only NFH positive neurites at > 200 μm distances from the cell body were analyzed using Fiji software.[47] Ostu thresholding was applied to generate a binary mask of the NFH signal, and PLA signal was then detected using the "Find Maxima…" function.

**Fluorescence recovery after photobleaching (FRAP)**. FRAP was used to test for axonal mRNA protein synthesis using diffusion-limited GFP$^{MYR}$ and mCherry$^{MYR}$ reporters as described with minor modifications[22]. In each case, DRG neurons were co-transfected with GFP$^{MYR}$5′/3′nrn1 + mCherry$^{MYR}$5′/3′gap43 or GFP$^{MYR}$5′/3′ impβ1 + mCherry$^{MYR}$5′/3′gap43 so that recovery of both reporters could be analyzed simultaneously. Cells were maintained at 37 °C, 5% CO$_2$ during imaging sequences. 488 nm and 514 nm laser lines on Leica SP8X confocal microscope were used to bleach GFP and mCherry signals, respectively (Argon laser at 70 % power, pulsed every 0.82 s for 80 frames). Pinhole was set to 3 Airy units to ensure full thickness bleaching and acquisition (63 × /1.4 NA oil immersion objective)[22]. Prior to photobleaching, neurons were imaged every 60 s for 2 min to acquire baseline fluorescence the region of interest (ROI; 15 % laser power, 498–530 nm for GFP and 565–597 nm for mCherry emissions, respectively). The same excitation and emission parameters were used to assess recovery over 15 min post-bleach with images acquired at 30 s intervals. To determine if fluorescence recovery in axons was from translation, cultures were treated with 150 μg/ml cycloheximide (Sigma) or 100 μm anisomycin (Sigma) for 30 min prior to photobleaching for GFP$^{MYR}$5′/3′nrn1 + mCherry$^{MYR}$5′/3′gap43and GFP$^{MYR}$5′/3′impβ1 + mCh$^{MYR}$5′/3′gap43 transfected DRGs, respectively[16,48]. For peptide treatments, G3BP1-mCh transfected DRG neurons were treated with 10 μM G3BP1 peptides after acquiring the baseline expression values. Photobleaching followed by analyses of recovery was performed after 30 min of peptide exposure.

For testing G3BP1 protein mobility in axons, DRG neurons were transfected with G3BP1$^{S149A}$-GFP or G3BP1$^{S149E}$-GFP and imaged as above but only the 488 nm laser was used for photobleaching (Argon laser at 70 % power, pulsed every 0.82 s for 80 frames).

Fluorescent intensities in the ROIs were calculated by the Leica LASX software. For normalizing across experiments, fluorescence intensity value at $t = 0$ min post-bleach from each image sequence was set as 0%. The percentage of fluorescence recovery at each time point after photobleaching was then calculated by normalizing relative to the pre-bleach fluorescence intensity (set at 100%)[49].

**Live cell imaging for G3BP1-mCherry granules**. DRG neurons were transfected with G3BP1-mCherry, and 36 h later distal axons were imaged using Leica SP8X confocal microscope with environmental chamber maintained at 37 °C, 5 % CO$_2$ (with 63 × /1.4 NA oil immersion objective). G3BP1-mCherry signals were imaged as single optical planes in the axon shaft every 2 s for 100 frames (at 540 nm excitation and 23 % white light laser power; 565–597 nm emission). To study the effect of the G3BP1 190–208 or 168–189 peptide, 10 μM FITC-conjugated peptide was added to the media and 15 min later imaging was continued.

For quantitation of G3BP1-mCherry aggregates density and size, a 100 μm of the axon shaft was considered ( ≥ 200 μm from cell body). Thresholding was applied to acquired image sequences using ImageJ to generate binary masks. ImageJ particle analyzer was used for analysis. G3BP1 aggregates with area ≥ 1 μm²

were considered as SG-like structures. For analyzing the G3BP1 aggregate velocity, ImageJ Trackmate plug-in was used[50].

**Puromycinylation assay**. To visualize newly synthesized proteins in cultured neurons, we used the Click-iT® Plus OPP Protein Synthesis Assay Kit per manufacturer's instructions (Invitrogen/Life Technologies). Briefly, 3 DIV cultures were incubated with 20 μM o-propargyl-puromycin (OPP) for 30 min at 37 °C. OPP-labeled proteins were detected by crosslinking with Alexa Fluor-594 picolyl azide molecule. Coverslips were then mounted with Prolong Gold Antifade (Invitrogen) and imaged with Leica DMI6000 epifluorescent microscope as above. ImageJ was used to quantify the Puromycinylation signals in distal axons and cell bodies.

**Immunoblotting**. For immunoblotting, protein lysates or immunoprecipitates were denatured by boiling in Laemmle sample buffer, fractionated by SDS-PAGE, and transferred to nitrocellulose membranes. Blots were blocked for 1 h at room temperature with 5% non-fat dry milk in Tris-buffered saline with 0.1% Tween 20 (TBST) for anti-tagBFP, -GAPDH and -G3BP1 antibodies; 5% BSA in TBST was used for blocking anti-G3BP1$^{PS149}$ antibody. Primary antibodies diluted in appropriate blocking buffer were added to the membranes and incubated overnight incubation at 4 °C with rocking. Primary antibodies consisted of: rabbit anti-G3BP1 (1:2000; Sigma), rabbit anti-G3BP1$^{PS149}$ (1:1000; Sigma), rabbit anti-TagBFP (1:2000; Evrogen), and rabbit anti-GAPDH (1;2000; CST). After washing in TBST, blots were incubated HRP-conjugated anti-rabbit IgG antibodies (1:5000; Jackson lab) diluted in blocking buffer for 1 h at room temperature. After washing signals were detected using ECL Prime$^{TM}$(GE Healthcare).

**Mass spectrometry by parallel reaction monitoring (PRM)**. Protein extraction was carried out according to the standard manufacturer's protocol using axoplasm samples suspended in 0.5 ml of TrIzol LS. Protein pellets were then reconstituted in urea, reduced, alkylated, digested with trypsin and desalted as previously described[51]. PRM was performed using nanoAcquity UPLC system (Waters) online with Q Exactive Plus mass spectrometer (Thermo-Fisher). Digested peptides were loaded at 0.5 μg per sample and separated by low pH, two-buffer reverse phase chromatography on a 200 cm monolithic silica-C18 column (GL Sciences, Japan) over a 6 h gradient as previously described[51]. Q Exactive Plus instrument was used in PRM mode with the following parameters: positive polarity, $R = 17,500$ at 200 $m/z$, AGC target 1e6, maximum IT 190 ms, MSX count 1, isolation window 3.0 $m/z$, NCE 35%. Unique previously described tryptic peptide DFFQSYGNVVELR from rat G3BP1 (Uniprot ID D3ZYS7) was targeted (as part of a set of 184 target peptides from 84 proteins with possible roles in axonal mRNA transport; subject of a separate study). PRM data analysis was performed using Skyline v. 3.5[52].

**RNA immunoprecipitation (RIP)**. HEK293T cells or DRG neurons were lysed in 100 mM KCl, 5 mM MgCl$_2$, 10 mM HEPES [pH 7.4], 1 mM DTT, and 0.5% NP-40 (RIP buffer) supplemented with 1 × protease inhibitor cocktail (Roche) and RNasin Plus (Invitrogen). Cells were passed through 25 Ga needle 5–7 times and cleared by centrifugation at 12,000×g for 20 min. Cleared lysates were pre-absorbed with Protein A-Dynabeads (Invitrogen) for 30 min. Supernatants were then incubated with primary antibodies for 3 h and then immunocomplexes precipitated with Protein G-Dynabeads (Invitrogen) for additional 2 h at 4 °C with rotation. Mouse anti-G3BP1 (5 μg, BD Biosciences) and rabbit anti-GFP (5 μg, Abcam) antibodies were used for immunoprecipitation. Beads were washed six times with cold RIP buffer. Bound RNAs were purified and analyzed by RTddPCR (see below).

**RNA isolation and PCR analyses**. RNA was isolated from immunoprecipitates and cultures using the RNeasy Microisolation kit (Qiagen). Fluorimetry with Ribogreen (Invitrogen) was used for RNA quantification. For analyses of total RNA levels and inputs for RIP analyses, RNA yields were normalized across samples prior to reverse transcription using Sensifast (Bioline). For RIP assays, an equal proportion of each RIP was used for reverse transcription with Sensifast. ddPCR products were detected using Evagreen or Taqman primer and probe sets (Biorad or Integrated DNA Tech; sequences available on request) and QX200$^{TM}$ droplet reader (Biorad). In GFP RIP experiment, B domain-BFP expression consistently increased G3BP1-GFP levels in the DRG neurons. So the level of mRNA precipitating with G3BP1-GFP was normalized to the G3BP1-GFP signals from immunoblotting across each sample in each experiment.

**Assessment of muscle reinnervation**. AAV5 encoding GFP or B domain-BFP was injected into the sciatic nerve near the sciatic nerve notch (left and right nerves, respectively). 7 d post-virus injections, bilateral sciatic nerve crushes were performed at mid-thigh level. The extent of innervation of the LG and TA muscles was evaluated using in vivo electromyography (EMG). For this, animals were anesthetized by IP injection with Ketamine HCl (80 mg/kg, Hospira) and AnaSed (10 mg/kg, LLOYD Laboratories) to achieve a surgical plane of general anesthesia; additional IP injections of Ketamine HCl (40 mg/kg) were given to maintain this plane throughout the experiment.

Bipolar fine wire EMG electrodes were constructed from insulated nickel alloy wire (California Fine Wire, Stablohm 800)[53]. The insulation over the distal 1 mm of

the tips was removed by scraping with a scalpel blade and the tips of the two wires were staggered by 1 mm. Electrodes were then placed into the lateral head of LG and the mid-belly of the TA muscles using a 25 Ga hypodermic needle. Once in place, the needle was removed and the wires were connected to the differential amplifiers. To stimulate the sciatic nerve, a small skin incision was made just inferior to the ischial tuberosity, exposing the sciatic nerve as it coursed between the gluteal and hamstring muscles, proximal to the crush injury site. A small rectangle of Parafilm® was wrapped loosely around the nerve and pierced by two unipolar needle electrodes (Neuroline monopolar, 28 G, Ambu/AS). The tips of the two needle electrodes were separated from each other by approximately 1 mm. Lead wires from the needles were connected to an optically isolated constant voltage stimulator under computer control.

Evoked EMG activity from LG and TA was then recorded after sciatic nerve stimulation. Stimulation and recording were controlled by a laboratory computer system running custom software written in Labview®. Ongoing EMG activity in the LG was sampled at 10 kHz; when the rectified and integrated voltage over a 20 ms period fell within a user-defined range, a 0.3 ms duration stimulus pulse was delivered to the nerve via the needle electrodes. Muscle activity was sampled from 20 ms prior to the stimulus until 100 ms after the stimulus and recorded to disc. Stimuli were delivered no more frequently than once every 3 s to avoid fatigue. A range of stimulus intensities was applied in each experiment to sample evoked muscle activity from sub-threshold to supramaximal. In a typical experiment approximately 200 stimulus presentations were studied. At the end of each experiment, all electrodes were removed and the skin incisions closed with sutures.

The recorded compound muscle action potentials ($M$ waves) in LG and TA evoked by sciatic nerve stimulation were analyzed off-line. The amplitude of the evoked $M$ waves was measured as the average rectified voltage within a defined time window after the stimulus application. In intact anesthetized animals, this window is 0.5–2.0 ms, as described[54]. After nerve crush, $M$ waves evoked from sciatic nerve stimulation are, by definition, generated by reinnervated muscle fibers. The latency and duration of these potentials are longer than those found in intact animals[54]. Thus, the time window used to measure the amplitude of the $M$ waves was adjusted to accommodate this change. Recordings were made from intact animals, immediately following and 1, 2, 4, 6, and 8 wk after nerve crush. At each time point, the amplitude of the largest evoked $M$ wave ($M_{max}$) was determined and scaled to $M_{max}$ recorded from that animal prior to nerve crush. Means of these scaled responses recorded from muscles in which motor neurons were induced to express B domain-BFP and those in which motor neurons expressed only GFP were compared at each time studied.

**Image analyses and processing.** For protein–protein and protein–mRNA colocalization, $xyz$ image sequences captured 100 μm segments of the axon shaft (separated from the cell body and growth cone by ≥ 200 μm) were deconvolved using Huygens HyVolution software. Colocalization was analyzed using ImageJ JACoP plug-in (https://imagej.nih.gov/ij/plugins/track/jacop.html) to calculate Pearson's coefficient. These coefficient calculations were independently validated with Volocity software (Perkin Elmer).

For analyses of protein levels in tissues, $z$ planes of the $xyz$ tile scans from 3–5 locations along each nerve section were analyzed using ImageJ. Colocalization plug-in was used to extract protein signals that overlap with axonal marker (NF) in each plane, with the extracted 'axon-only' signal projected as a separate channel[55]. For calculating axonal G3BP1 aggregate and G3BP1$^{PS149}$ signal intensities, absolute signal intensity was quantified in each $xy$ plane of the 'Colocalization' extracted images for axonal only G3BP1 and G3BP1$^{PS149}$ using ImageJ. Protein signal intensities across the individual $xy$ planes were then normalized to NF immunoreactivity area. The relative protein signal intensity was averaged for all image locations in each biological replicate.

For neurite outgrowth, images from 60 h DRG cultures were analyzed for neurite outgrowth using WIS-Neuromath[56]. Axon morphology was visualized using GFP and/or NF immunofluorescence as described[48]. Differentiated hiPSC neuron image acquisition and neurite length quantification was performed using Arrayscan XTI (Thermo Fisher).

To assess regeneration in vivo, tile scans of NF-stained nerve sections were post-processed by Straighten plug-in for ImageJ (http://imagej.nih.gov/ij/). NF positive axon profiles were then counted in 30 μm bins at 0.3 mm intervals distal from crush site. Number of axon profiles present in the proximal crush site was treated as the baseline, and values from the distal bins were normalized to this to calculate the percentage of regenerating axons.

**Statistical analyses.** Kaleidagraph (Synergy), Prism (GraphPad), and Excel (Microsoft) software packages were used for statistical analyses. One-way ANOVA was used to compare means of independent groups and Student's $t$-test was used to compare smaller sample sizes of the in vivo analyses. $p$ values of ≤ 0.05 were considered as statistically significant. For statistical analyses of Pearson's coefficients, Fishers Z-transformation was used to compare: G3BP1 + HuR, FXR1, and FMRP colocalization vs. DCP1a + XRN1 coefficients and G3BP1 + DCP1a and XRN1 vs. DCP1a + XRN1 coefficients.

**Data availability.** The data that support the findings of this study are available from the corresponding author upon reasonable request.

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

## Acknowledgements

This work was supported by grant funds from National Institutes of Health (R01-NS041596 and R01-NS089633 to J.L.T. and NS057190 to A.W.E.), Department of Defense—Congressionally Mandated Research Program (W81XWH-2013-1-308 OR120042 to J.L.T. and M.F.), and the Dr. Miriam and Sheldon G. Adelson Medical Research Foundation (to A.L.B., C.J.W., J.L.T., and M.F.). MS experiments were performed at the Biomedical Mass Spectrometry Resource at UCSF supported by funding from the Howard Hughes Medical Institute (to A.L.B.). The Histochemistry and Tissue Processing Core of Nemours/Alfred I. duPont Hospital performed tissue processing for Children. M.F. is the incumbent of the Chaya Professorial Chair in Molecular Neuroscience at the Weizmann Institute of Science, and J.L.T. is the incumbent of the SmartState Chair in Childhood Neurotherapeutics at the University of South Carolina.

## Author contributions

P.K.S., A.W.E. and J.L.T. designed the study. Experiments were performed by P.K.S., S.J.L., P.B.J., S.A., A.N.K., S.M.R., T.S., B.S., E.E.T., T.S.H. and A.U., A.L.B., C.J.W., M.F., and A.W.E. provided critical reagents, advanced instrumentation, experimental models, and expert guidance. E.A.P. provided statistical analyses. J.L.T. and P.K.S. wrote the manuscript. All authors commented on the manuscript draft and approved the final version.

## Additional information

**Competing interests:** The authors declare no competing interests.

