## [Peer Review File · Nature Communications]

Reviewers' comments:

Reviewer #1 (Remarks to the Author):

Cytoplasmic stress granules have received much attention recently, as they are believed to give rise to aggregates containing TDP-43 and TIA1 in neurodegenerative disorders. However, the physiological function of these granules is not well understood. In particular, it is not well characterized whether stress granules form in axons, and what the function of these structures in axons could be. This study focuses on the role of G3BP1 in axons, its regulation after axotomy and how it could modulate regeneration. The authors show that G3BP1 is present in axons in stress granule-like aggregates, that G3BP1 levels decrease after nerve lesion and increase in regenerating axons. The authors also show that G3BP1 is phosphorylated in regenerating axons which apparently inhibits stress granule formation. The authors then show that some candidate transcripts known to be translated in regenerating axons, in particular neuritin-1 (Nrn1) is translated in a G3BP1-dependent manner and that an acidic domain located between AA141-220 of G3BP1 mediates the effects to increase axonal mRNA translation and inhibit stress granule formation. This paper also contains some preliminary data that a peptide corresponding to this domain positively affects axon regeneration after nerve lesion. These are interesting and novel data on how stress granules can modulate protein synthesis in axon regeneration. However, this paper also contains some technical flaws that need to be amended.

1. A G3BP1 knockout mouse has been generated and investigated in the lab of J. Tazi. G3BP1-deficient mice show defects in motor function, short-term plasticity, intracellular calcium homeostasis, which might be relevant in the context of this study. This paper (Martin et al., J. Neurochem. 125: 175-184, 2013) should be cited and discussed. It would be interesting to see whether nerve regeneration is altered after crush lesion in G3BP1-deficient mice.
2. The data shown in Fig. 1a on the axonal location of G3BP1 are fully not convincing. First, additional typical markers for stress granules such as TIA1 and Caprin-1 should be included in this analysis. These two proteins represent classic interaction partner of G3BP1. Such data would strengthen this part of the study.
3. The data shown in Fig. 1e are not convincing. G3BP1 immunoreactivity is hard to detect, and it is not clear whether it is present diffusely in axons or in stress granule-like particles. This would be particularly relevant to understand whether such stress granule-like structures form directly after crush within the first 5 days and how they resolve when regeneration starts at 7 days after lesion. Therefore, different time points should be included in this analysis, and additional markers such as TIA1 included to distinguish diffuse G3BP1 from stress granule-like aggregates. A control for the specificity of the G3BP1 staining needs to be included to proof that the diffuse G3BP1 staining is specific and does

not represent background staining. The Western blot shown in Ext. Data Fig. 1b are not convincing, because a knockdown control is missing.

4. The authors show by immunohistochemistry in Fig. 1f-h and by mass spectrometry that G3BP1 levels decline after injury. Fig. 1g suggests a dramatic effect with a reduction by about 80 %. However, this is not reflected by the mass spectrometry data in Ext. Data Fig. 1a. These data need to be confirmed by Western blot analysis of nerve extracts. The n for independent experiments in Ext. Data Fig. 1a should be given, and the data should be presented in a different way, with individual data points instead of whiskers. Which of these data points represent significant deviations from control?

5. The data shown in Fig. 2f and g that phosphor-S149-G3BP1 is increased in distal segments of growing axons are highly interesting. The authors should discuss the mechanisms that could underlie the differential phosphorylation in distal vs. proximal axonal segments. What is the kinase responsible for this effect, and how is it regulated ?

6. The experiments shown in Fig. 4d and e are designed to show that the 190-208 peptide can increase axon regeneration after crush lesion in adult rats. These are important and key data within this study, but they are technically not convincing. First, the variation as shown by the whiskers is extremely high for data points at 6 and 8 weeks. This is probably due to the low numbers of animals used (n=6). Therefore, variations should not be shown as whiskers, but individual data points. Are these data normally distributed?

For the tibialis anterior muscle, only the M-response at 4 weeks seems to be significant, but not at 6 and 8 weeks. This is also true for the M-response at 8 weeks after nerve lesion in the gastrocnemius muscle. If this peptide enhances regeneration, then more lesioned axons should have successfully regrown to the muscle at 6-8 weeks, and the difference in M-response should also be seen at 8 weeks after lesion. This study would be much more convincing if these crush lesions were done in the G3BP1 knockout mice described by the Tazi group.

Reviewer #2 (Remarks to the Author):

This is an important paper which clarifies a general mechanism for the switching on and off of mRNA translation in sensory axons in response to damage. It has been known for some time that local translation is necessary for efficient regeneration of sensory axons, but the mechanism that regulates it has not been clear. The paper takes an important step forward in showing that sequestration of mRNAs into stress granules can inhibit translation, while release of the RNAs after damage can enable both translation and regeneration. Phosphorylation of the RNA binding protein G3BP1 is the control mechanism. The paper contains a large amount of data which cannot be presented in detail because of the length limit. However, the data that is shown is convincing and consistent, and the work comes from leaders in this field.

1. Page 4. Intra-axonal G3BP is found to be lowered proximal to a nerve crush. I assume that it is also lowered in the regenerating axons distal to the crush, but readers could misinterpret this statement as excluding the regenerating axons. The regenerating axons

are thin and hard to stain and co-localize, but if there is a picture of the regenerating axons it would be good to show it.

2. The control of stress granules by phosphorylation of G3BP1 is a key finding, and could be treatment target. Did the expression of the phospho-mutants have any effect on granule formation, and any effect on translation or regeneration? Or is there too much endogenous protein?

3. There is an inconsistency on page 6. In the first section overexpression of G3BP1 decreased protein translation, but in the second section overexpression had no effect on axon growth- indicating that the molecule is at saturating levels. Does expression of the BFP form make it possible to estimate endogenous vs overexpressed levels?

4. By localizing a small recognition site on the protein, and finding mRNAs that either do or do not associate with it, does a recognition motif on the RNA come out?

5. Is there any information about which enzymes might modulate phosphorylation of G3BP?

Reviewer #3 (Remarks to the Author):

Sahoo et al. investigate how interfering with G3BP1 function affects the growth and regeneration of peripheral axons. They suggest that G3BP1 sequesters mRNAs in stress granules (SGs) and thereby suppresses translation in axons to limit their growth. The expression of the G3BP1 B domain or the incubation with Tat-fusion peptides derived from it modulates axonal translation and enhances regeneration. The authors apparently conclude (but do not discuss in detail) that axonal injury results in G3BP1 phosphorylation and disassembly of SGs, which increases the translation of proteins like Nrn1. Results and Discussion are very condensed and brief, it is often difficult to deduce the details of the experiment and the conclusions drawn from the results. The authors use reporters for three transcripts (Nrn1, Imp β 1 and Gap43) to analyze how G3BP1 influences translation but do not test if the translation of the corresponding endogenous mRNAs is affected in the same way. The effects of full-length G3BP1 and G3BP1 domains on the translation reporters and axon growth correlate only partially (see point 4 below). It is suggested that the increased translation of Nrn1 promotes axon growth but this is also not shown directly. The results presented in the manuscript indicate that the G3BP1 B domain accelerates regeneration but it appears premature to attribute this to changes in the translation of specific transcripts.

Additional points

1) The antibody to detect G3BP1 phosphorylation at S149 is not validated for immunofluorescence. According to the manufacturer and Ext. Data Fig. 1b the antibody was tested only by Western blot and detects multiple bands. The specificity of the G3BP1 antibodies has to be validated also for immunofluorescence with G3BP1-deficient neurons.

2) The Pearson's coefficient for the co-localization of G3BP1 and SG markers is low and does not differ much from that for processing bodies (PBs) (SGs: 0.23 vs. HuR, PBs: 0.15 vs. DCP1A), which raises the question if the focus on SGs is justified.

3) Reporters with the 5' and 3' UTR of Nrn1, Imp β 1 and Gap43 were used to quantify effects on local translation by FRAP. In Fig. 3d (and Ext. Data movies) fluorescence recovery appears to be high when there are also strong signals for the reporter in the axon shaft outside the ROI while those with low signals in the shaft also show low recovery. How can the authors exclude that the recovery depends to a large extent on the diffusion or transport of MYR-GFP from the axon shaft?

4) Expression of full-length G3BP1 reduces translation of the Nrn1 and Imp β 1 reporters but does not affect axon length. The BFP control is missing from Ext. Data Fig. 2b - c and the effect of the B domain is compared only to that of the D domain. When comparing these results to the BFP control from Fig. 3e, g the B domain does not appear to have a significant effect and the D domain represses translation to a similar extent as full-length G3BP1 at least for Nrn1. In contrast, the D domain but not full-length G3BP1 reduces and the B domain increases axon length (Ext. Data Fig. 3a). The absence of an effect on axon growth is explained by saturating levels of G3BP1 (p. 6) but this should also apply to the effect on translation.

5) Most of the experiments are based on the overexpression of G3BP1 domains or treatment with Tat-fusion proteins while only axon length is quantified for the knockdown. The effects of a knockdown on translation and SGs should also be tested.

Minor points

1) The references (6, 7) for "The RasGAP SH3 domain binding protein 1 (G3BP1) interacts with the 48S pre-initiation complex when translation is stalled, and it assembles SGs by virtue of its NTF2-like domain" (p. 3) appear to be wrong.

2) The results in Ext. Data should be arranged in the same order as they are mentioned in the results (e.g. p. 5, Ext. Data Fig. S5B).

3) Fig. 5f: the cultures are incubated with the Tat-fusion peptide for 15 min. It is surprising that this is sufficient to have an effect. As a control no peptide is added. Instead an inactive peptide has to be used as control.

NCOMMS-17-33728 – Response to Reviewers

We appreciate the reviewers' careful assessment of our previous submission. We have made substantial changes to the revised manuscript, adding new experiments and editing the text to specifically address issues raised at review. We think that these revisions have strengthened the manuscript and provide further support for our conclusion that stress granule-like structures are used to store axonal mRNAs and thereby attenuate axon growth rates. We hope that the reviewers and editor will agree. We outline the changes in the sections below with regards to the individual reviewers' comments (in red font).

Reviewer #1:

Cytoplasmic stress granules have received much attention recently, as they are believed to give rise to aggregates containing TDP-43 and TIA1 in neurodegenerative disorders. However, the physiological function of these granules is not well understood. In particular, it is not well characterized whether stress granules form in axons, and what the function of these structures in axons could be. This study focuses on the role of G3BP1 in axons, its regulation after axotomy and how it could modulate regeneration. The authors show that G3BP1 is present in axons in stress granule-like aggregates, that G3BP1 levels decrease after nerve lesion and increase in regenerating axons. The authors also show that G3BP1 is phosphorylated in regenerating axons which apparently inhibits stress granule formation. The authors then show that some candidate transcripts known to be translated in regenerating axons, in particular neuritin-1 (*Nrn1*) is translated in a G3BP1-dependent manner and that an acidic domain located between AA141-220 of G3BP1 mediates the effects to increase axonal mRNA translation and inhibit stress granule formation. This paper also contains some preliminary data that a peptide corresponding to this domain positively affects axon regeneration after nerve lesion. These are interesting and novel data on how stress granules can modulate protein synthesis in axon regeneration. However, this paper also contains some technical flaws that need to be amended.

We thank the reviewer for noting the interesting and novel data in our manuscript. We hope the reviewer will regard additional data presented here as more mature, including those for the cell permeable 190-208 peptide that is a potentially translatable reagent.

1. A G3BP1 knockout mouse has been generated and investigated in the lab of J. Tazi. G3BP1-deficient mice show defects in motor function, short-term plasticity, intracellular calcium homeostasis, which might be relevant in the context of this study. This paper (Martin et al., *J. Neurochem.* 125: 175-184, 2013) should be cited and discussed. It would be interesting to see whether nerve regeneration is altered after crush lesion in G3BP1-deficient mice.

Although we agree that such approaches will be interesting to pursue in the future, the currently available G3BP1 knockout mouse models do not allow injury and regeneration analyses. The originally published knockout was found to be embryonic lethal on the 129/Sv genetic background (Zekri et al., 2005, *Mol. Cell. Biol.* 25, 8703; Martin et al., 2013, *J. Neurochem.* 125, 175). This was partly circumvented by repetitive backcrossing to Balb/c background for 10 generations, and then maintaining the mice as mixed 50:50 129/Sv:Balb/c. However over 80% of these mixed background knockouts do not survive to adulthood (2 months of age), and those that do survive “*demonstrate a striking akinesia; most KO mice dragged their limbs behind them when walking*” (Martin et al., 2013). Hence, unfortunately these mice cannot be used for nerve injury and regeneration analyses. On the other hand, the acute knockdown and peptide perturbant approaches we have used (**Fig. 4c-d and**

Ext. Data Fig. 5) enabled specific testing of the role of G3BP1 in adult injury, without confounding issues that might arise in general knockout models as noted above.

2. The data shown in Fig. 1a on the axonal location of G3BP1 are fully not convincing. First, additional typical markers for stress granules such as TIA1 and Caprin-1 should be included in this analysis. These two proteins represent classic interaction partner of G3BP1. Such data would strengthen this part of the study.

We provide new images for **Fig. 1a**. Additionally, **Fig. 1b-d** shows colocalization of axonal G3BP1 with other stress granule markers in DRG cultures and **Fig. 1e** shows colocalization of axonal G3BP1 with TIA1 in sciatic nerve in vivo.

3. The data shown in Fig. 1e are not convincing. G3BP1 immunoreactivity is hard to detect, and it is not clear whether it is present diffusely in axons or in stress granule-like particles. This would be particularly relevant to understand whether such stress granule-like structures form directly after crush within the first 5 days and how they resolve when regeneration starts at 7 days after lesion. Therefore, different time points should be included in this analysis, and additional markers such as TIA1 included to distinguish diffuse G3BP1 from stress granule-like aggregates. A control for the specificity of the G3BP1 staining needs to be included to proof that the diffuse G3BP1 staining is specific and does not represent background staining. The Western blot shown in Ext. Data Fig. 1b are not convincing, because a knockdown control is missing.

We have added knockdown controls for the western blots as **Ext. Data Fig. 1b**. Note that only a single band is detected with this G3BP1 antibody. We have added new figures for the in vivo G3BP1 imaging in naïve and regenerating sciatic nerve with TIA1 colabeling as requested (**Fig. 1e**). Additionally, we show that the aggregation of TIA1 in axons near perfectly overlaps with that of axonal G3BP1 aggregation (**Fig. 1f**).

4. The authors show by immunohistochemistry in Fig. 1f-h and by mass spectrometry that G3BP1 levels decline after injury. Fig. 1g suggests a dramatic effect with a reduction by about 80 %. However, this is not reflected by the mass spectrometry data in Ext. Data Fig. 1a. These data need to be confirmed by Western blot analysis of nerve extracts. The n for independent experiments in Ext. Data Fig. 1a should be given, and the data should be presented in a different way, with individual data points instead of whiskers. Which of these data points represent significant deviations from control?

The dramatic reductions for G3BP1 immunofluorescent signals shown in original Fig. 1g (now replaced with **Fig. 1f**) and images are based on imaging parameters to detecting only the granular G3BP1 signal (**Fig. 1e**). If we overexpose, we do see diffuse G3BP1 immunoreactivity separated from the (highly overexposed) G3BP1 granule-like structures (this is now noted in the text). We fully agree that the mass spectrometry data would contradict these immunofluorescence data if the images displayed total G3BP1 rather than aggregated G3BP1. We have emphasized in the manuscript that the mass spectrometry data were repeated across three replicates and that there is substantial variability for the fold change values relative to naïve animals. These data required 3 animals per time point, taking approximately 3 cm of proximal sciatic nerve for axoplasm preparation (from crush site to -3 cm for injured nerve). To better address the axonal G3BP1 levels, we performed immunoblotting for G3BP1 on axoplasm preparations for nerve segments taken at 0 to -1 and -1 to -2 cm proximal to the crush site (as compared with 0 to -3 cm segment used for mass spectrometry) and a corresponding segment of uninjured sciatic nerve. The -1 to -2 cm segment of

the injured nerve is indistinguishable from naïve while the 0 to -1 cm shows clear decrease in axoplasm G3BP1 signals (**Ext. Data Fig. 1d**).

5. The data shown in Fig. 2f and g that phosphor-S149-G3BP1 is increased in distal segments of growing axons are highly interesting. The authors should discuss the mechanisms that could underlie the differential phosphorylation in distal vs. proximal axonal segments. What is the kinase responsible for this effect, and how is it regulated?

We agree with the reviewer that the increase in G3BP1^{PS149} is extremely interesting and are now pursuing the protein kinase. We have added text to indicate that activation of a protein kinase in axons after injury could account for the increase in axonal phospho-G3BP1 and noted that both CK2 α and AKT have been reported to phosphorylate G3BP1 in other cellular systems (Kwok et al., 2017, Cell Mol Life Sci 74, 3613-30; Reineke et al., 2017, Mol Cell Biol 37, issue 4).

[redacted]

This issue will clearly need substantial additional work in the future, and we hope that the reviewers and editor will agree that the current study on G3BP1 aggregation stands on its own, and the mechanisms underlying this post-translational modification of G3BP1 are beyond the scope of the present manuscript.

[redacted]

6. The experiments shown in Fig. 4d and e are designed to show that the 190-208 peptide can increase axon regeneration after crush lesion in adult rats. These are important and key data within this study, but they are technically not convincing. First, the variation as shown by the whiskers is extremely high

for data points at 6 and 8 weeks. This is probably due to the low numbers of animals used (n=6). Therefore, variations should not be shown as whiskers, but individual data points. Are these data normally distributed?

For the tibialis anterior muscle, only the M-response at 4 weeks seems to be significant, but not at 6 and 8 weeks. This is also true for the M-response at 8 weeks after nerve lesion in the gastrocnemius muscle. If this peptide enhances regeneration, then more lesioned axons should have successfully regrown to the muscle at 6-8 weeks, and the difference in M-response should also be seen at 8 weeks after lesion. This study would be much more convincing if these crush lesions were done in the G3BP1 knockout mice described by the Tazi group.

Please note that the data in Fig. 4d-e of the original manuscript used AAV5 expressing G3BP1 domains rather than the cell permeable peptide. To avoid this confusion, we have rearranged the order of Fig. 4 so that the protein expression experiments are now shown as **Fig. 4c-d** with the peptide treatment data in **Fig. 4e** (also note that **Ext. Data Fig. 6** is now only data from the cell permeable peptides).

We have now included the individual data points for the new **Fig. 4d** where compound muscle action potentials were used to evaluate muscle reinnervation. The data are normally distributed and give statistically significant differences between control and G3BP1 B domain expressing animals. While adding additional animals might indeed reduce the error bars, it is difficult to justify such a request to our institute's IACUC given the fact that this experiment has already reached statistical significance. Moreover, there is also variation in transduction efficiency between animals that introduce greater error than we see in vitro. We hope that the reviewer will agree that the combination of the data in **Fig. 4c-d** and **Ext. Data Fig. 5** provide a convincing case for growth promotion by inhibition of G3BP1 with expression of the G3BP1 B domain.

For **Fig. 4c**, we tried putting all of the individual data points on a single graph, but it was a terribly busy graph and, we believe, less informative than the mean \pm SEM. We include the individual data points on graphs below as **Reviewers' Fig. 2**, that shows the B domain transduced animals clearly segregate from the G3BP1-BFP, G3BP1 D domain-BFP, and GFP expressing animals.

Reviewers' Figure 2: Data from Fig. 4c plotted from the 4 groups to show data points for individual animals. Black horizontal lines show the mean with SEM indicated by vertical capped lines. Dashed green line indicates the 50% value relative to crush site.

With the mid-thigh sciatic nerve crush injury used here, we consistently begin to see recovery of hind limb function by 4-5 weeks post-injury in untreated rats based on hind paw use. Consequently, it is expected that controls will show the same degree of reinnervation over time compared to the treated animals. We used the C-MAP analyses here as a clinically relevant and quantifiable measure of functional reinnervation. We interpret the data shown in **Fig. 4d** and **Ext. Data Fig. 5c** as showing

evidence for accelerated axon regeneration, so the controls 'catching up' is to be expected given the spontaneous regeneration of PNS axons. With the smaller innervation mass of and distance to regenerate to the tibialis anterior, it is not surprising that the control animals 'caught up' sooner with the G3BP1 B domain transduced animals for this muscle compared to the larger gastrocnemius. We have noted these points in the text, and emphasize in the discussion that the acceleration brought by targeting G3BP1 function in axons represents a significant advance considering the 1-2 mm/day rate regeneration in the PNS.

Reviewer #2:

This is an important paper which clarifies a general mechanism for the switching on and off of mRNA translation in sensory axons in response to damage. It has been known for some time that local translation is necessary for efficient regeneration of sensory axons, but the mechanism that regulates it has not been clear. The paper takes an important step forward in showing that sequestration of mRNAs into stress granules can inhibit translation, while release of the RNAs after damage can enable both translation and regeneration. Phosphorylation of the RNA binding protein G3BP1 is the control mechanism. The paper contains a large amount of data which cannot be presented in detail because of the length limit. However, the data that is shown is convincing and consistent, and the work comes from leaders in this field.

We appreciate the reviewer's encouraging comments on the impact of this work. We agree that we have a large amount of data here. Considering that there is now even more data with the revisions and the Journal's length limitations, we include some data for reviewers' use to address detail issues in this response (*see Reviewers Figs. 1-4*).

1. Page 4. Intra-axonal G3BP is found to be lowered proximal to a nerve crush. I assume that it is also lowered in the regenerating axons distal to the crush, but readers could misinterpret this statement as excluding the regenerating axons. The regenerating axons are thin and hard to stain and co-localize, but if there is a picture of the regenerating axons it would be good to show it.

We have provided new images to show G3BP1 aggregates in the distal regenerating axons as **Ext. Data Fig. 1a**. As the reviewer indicated, these most distal axons are very thin and quite difficult to image, but with deconvolution of Z stacks, we were able to visualize those distal axon tips (or close to the tip).

2. The control of stress granules by phosphorylation of G3BP1 is a key finding, and could be treatment target. Did the expression of the phospho-mutants have any effect on granule formation, and any effect on translation or regeneration? Or is there too much endogenous protein?

We have included additional data to show the effect of G3BP1^{S159E} mutant on endogenous stress granule-like aggregates as **Ext. Data Fig. 2b-c**. Additionally, we show that expression of the G3BP1 phosphomimetic protein in DRGs does not affect axon growth (**Ext. Data Fig. 4b**). As the reviewer suggests, we think that these and the wild-type G3BP1 overexpression data point to saturating levels of endogenous G3BP1 – also as we note below in **Reviewers' Fig. 3**, the overexpressed G3BP1-BFP is only modestly changing overall G3BP1 levels. Consistent with the notion of G3BP1 being at saturating levels, knockdown of G3BP1 increases axon growth and intra-axonal translation (**Ext. Data Fig. 4c-d and Fig. 5c**), but overexpression of G3BP1 has little effect on axon outgrowth (**Fig. 4b and Ext. Data Fig. 4b**). Nonetheless, we do see moderate decreases in axonal Nrn1 and Impβ1 proteins

on overexpression of G3BP1, so it has effects that are probably not sufficient to affect axon outgrowth and overall translation. As we note in the revised discussion, it will be extremely interesting to determine the RNA interactome of G3BP1 and axonal SG-like structures under these conditions in future work.

3. There is an inconsistency on page 6. In the first section overexpression of G3BP1 decreased protein translation, but in the second section overexpression had no effect on axon growth- indicating that the molecule is at saturating levels. Does expression of the BFP form make it possible to estimate endogenous vs overexpressed levels?

We include immunoblots to show the levels of transfected G3BP1 protein vs. endogenous G3BP1 protein in the DRG cultures (*see Reviewers' Fig. 3*).

Reviewers' Figure 3: Immunoblots for endogenous G3BP1 and BFP-tagged G3BP1 levels in three separate transfections of primary DRG neurons. Endogenous G3BP1 is expressed at high levels compared to the G3BP1-BFP. This likely accounts for the lack any significant effect for G3BP1 overexpression on total protein synthesis in axons and axon growth shown in Figs. 5b & Ext. Data Fig. 4d of the revised manuscript.

4. By localizing a small recognition site on the protein, and finding mRNAs that either do or do not associate with it, does a recognition motif on the RNA come out?

This is an interesting possibility that will require significant new studies, well beyond the scope of the current manuscript. G3BP1 does have RNA binding activity, but so do many SG proteins, including TIA1, HuR, FMRP, and FXR1 included in this manuscript. Addressing this issue will require a sustained series of RNA-Seq, CLIP-Seq, and RIP-Seq approaches to sort out. Moreover, given that the interactions are occurring in the axons, we will need to perform these studies on isolated axons, which will be a highly challenging undertaking.

5. Is there any information about which enzymes might modulate phosphorylation of G3BP?

As noted above for Reviewer # 1, issue 5, we are highly interested in the kinase. CK2 and AKT have been linked to G3BP1 phosphorylation. We have initial data that CK2 may be involved (*see Reviewers' Fig. 1 above*). The protein is detected in PNS axons and its levels increase after injury, but as noted in the response to Reviewer 1 above, a comprehensive assessment of this issue is beyond the scope of the present study. We show that expression of the G3BP1 B domain and treatment with the potentially clinically relevant cell permeable peptide trigger disassembly of axonal SG-like structures, increase intra-axonal protein synthesis, and increase PNS and CNS axon growth. We hope that reviewers and the editor will agree that these exciting findings warrant publication in their present form.

Reviewer #3:

Sahoo et al. investigate how interfering with G3BP1 function affects the growth and regeneration of peripheral axons. They suggest that G3BP1 sequesters mRNAs in stress granules (SGs) and thereby suppresses translation in axons to limit their growth. The expression of the G3BP1 B domain or the incubation with Tat-fusion peptides derived from it modulates axonal translation and enhances regeneration. The authors apparently conclude (but do not discuss in detail) that axonal injury results in G3BP1 phosphorylation and disassembly of SGs, which increases the translation of proteins like *Nrn1*. Results and Discussion are very condensed and brief, it is often difficult to deduce the details of the experiment and the conclusions drawn from the results. The authors use reporters for three transcripts (*Nrn1*, *Impβ1* and *Gap43*) to analyze how G3BP1 influences translation but do not test if the translation of the corresponding endogenous mRNAs is affected in the same way.

We now include data showing that the endogenous *Nrn1* and *Impβ1* mRNAs, but not *Gap43* mRNA, are affected by manipulation of G3BP1 (**Fig. 5d and Ext. Data Fig. 7c**). Interestingly, axonal levels of both *Nrn1* and *Impβ1* proteins decrease on overexpression of G3BP1, but only *Nrn1* protein increases when the G3BP1 B domain is introduced. This is consistent with FRAP data presented in **Fig. 6c**, where overexpression of G3BP1 decreases axonal translation of both GFP^{MYR}5'/3'*nrn1* and GFP^{MYR}5'/3'*impb1*, but this only fully reversed for GFP^{MYR}5'/3'*nrn1*.

We have expanded the discussion to clarify these issues.

The effects of full-length G3BP1 and G3BP1 domains on the translation reporters and axon growth correlate only partially (see point 4 below). It is suggested that the increased translation of *Nrn1* promotes axon growth but this is also not shown directly. The results presented in the manuscript indicate that the G3BP1 B domain accelerates regeneration but it appears premature to attribute this to changes in the translation of specific transcripts.

Though increased axon growth with overexpression of axonally-targeted *Nrn1* mRNA was published by us (see *Fig. 4 in Merianda et al., 2013, J Neurosci 33, 13735*), we enthusiastically agree with the reviewer's point that it is premature to mechanistically contribute growth promotion with G3BP1 inhibition here to translation of specific axonal mRNAs. *Nrn1*, *Impβ1* and *Gap43* have proven useful tools to show translational regulation and specificity for G3BP1 inhibition and we do not conclude this in the manuscript. We have now further revised the discussion to emphasize this point. Clearly more work will be needed to address which mRNAs account for the growth-promoting effects of G3BP1 B domain and 190-208 peptide.

Additional points

1) The antibody to detect G3BP1 phosphorylation at S149 is not validated for immunofluorescence. According to the manufacturer and Ext. Data Fig. 1b the antibody was tested only by Western blot and detects multiple bands. The specificity of the G3BP1 antibodies has to be validated also for immunofluorescence with G3BP1-deficient neurons.

The reviewer is correct the manufacturer had not validated the anti-G3BP1^{PS149} antibody for immunofluorescence. By our immunostaining, the axonal signals for anti-G3BP1^{PS149} antibody are increased when the axonal aggregates seen with anti-pan G3BP1 decrease (**Fig. 2d-e vs. Fig 1e-f, and**

Fig. 2f-g). Ext. Data Fig. 1b shows that the anti-G3BP1^{PS149} (as well as anti-G3BP1) detects a single band by immunoblotting that is near completely lost in G3BP1 depleted neurons. We have also tested this antibody for immunofluorescence in G3BP1-depleted DRGs as suggested. This is now included in the methods section (signal was reduced). There are relatively few publications for this phospho-G3BP1 antibody with all focusing on non-neuronal systems (mostly homogenous cell lines), hence we are indeed most likely the first to show immunofluorescence for phospho-G3BP1.

2) The Pearson's coefficient for the co-localization of G3BP1 and SG markers is low and does not differ much from that for processing bodies (PBs) (SGs: 0.23 vs. HuR, PBs: 0.15 vs. DCP1A), which raises the question if the focus on SGs is justified.

We addressed these concerns by re-analyzing the data with help of an expert statistician, who is now also included as an author (Edsel A. Pena). With Professor Pena's guidance, we applied a Fisher's Z-transformation to statistically compare the Pearson's coefficients population data for the intra-axonal G3BP1 colocalization SG markers (HuR, FMRP, FXR1) vs. G3BP1 colocalization with PB markers (DCP1a, Xrn1) vs. Dcp1a colocalization with Xrn1. **Fig. 1c** shows that the population of Pearson's coefficient data for G3BP1 + SG markers is statistically greater than G3BP1 + PB markers and PB markers show statistically higher colocalization with one another than with G3BP1.

3) Reporters with the 5' and 3' UTR of *Nrn1*, *Impβ1* and *Gap43* were used to quantify effects on local translation by FRAP. In Fig. 3d (and Ext. Data movies) fluorescence recovery appears to be high when there are also strong signals for the reporter in the axon shaft outside the ROI while those with low signals in the shaft also show low recovery. How can the authors exclude that the recovery depends to a large extent on the diffusion or transport of MYR-GFP from the axon shaft?

Please note that others groups have consistently published this method for visualizing and quantifying localized protein synthesis (e.g., see: Science [2004] 304, 1979 from Erin Schuman; Nature [2006] 439, 283 from Michael E. Greenberg; Cell [2016] 166, 181 from Christine Holt; and J Cell Biol [2017] 217, 793 from Michael Sendtner). In our first publication with this FRAP method (Yudin et al., 2008, Neuron 59, 241-52), we were concerned over the possibility of diffusion from proximal segments. In the Yudin et al. paper, we used FRAP on isolated axons and a double photobleach method where recovery was monitored in the distal axon while the immediately proximal segment was continually photobleached to remove any diffusing GFP^{MYR} (see Figs. 4E and S6B-C in Yudin et al., 2008). To more directly address the reviewer's point, we have gone back to the original image sequences and quantified the GFP^{MYR} and mCherry^{MYR} signals in the proximal axon shafts over the FRAP time course. There are no significant changes in fluorescence over the course of the photobleaching and recovery experiments (see Reviewers' Fig. 4). We hope this fully addresses the reviewer's concerns, and we have cited the above methodological references in the manuscript.

Reviewers' Figure 4: Analyses of fluorescent recovery in axon shafts proximal to bleached ROI for image sequences from Fig. 3e-g are shown. There is no significant change in fluorescence in this proximal axon segment over the course of the photorecovery period.

4) Expression of full-length G3BP1 reduces translation of the Nrn1 and Impβ1 reporters but does not affect axon length. The BFP control is missing from Ext. Data Fig. 2b - c and the effect of the B domain is compared only to that of the D domain. When comparing these results to the BFP control from Fig. 3e, g the B domain does not appear to have a significant effect and the D domain represses translation to a similar extent as full-length G3BP1 at least for Nrn1. In contrast, the D domain but not full-length G3BP1 reduces and the B domain increases axon length (Ext. Data Fig. 3a). The absence of an effect on axon growth is explained by saturating levels of G3BP1 (p. 6) but this should also apply to the effect on translation.

We omitted the BFP controls as the FRAP sequences with all the data were very difficult to visualize. For this revision, we have specifically included new data for overall axonal protein synthesis in axons vs. cell bodies by puromycinylation (**Fig. 5a-c, Fig. 6a-b, and Ext. Data Fig. 7a**) and effects of full length G3BP1 vs. G3BP1 B domain expression on axonal levels of endogenous Nrn1, Impβ1 and Gap43 proteins (**Fig. 5d**). The data are clear and consistent. Additionally, we have included axonal FRAP data for GFP^{MYR5'/3'}nrrn1, GFP^{MYR5'/3'}impβ1, and mCh^{MYR5'/3'}gap43 in G3BP1-BFP expressing neurons ± 190-208 peptide as **Fig. 6c**. The 190-208 peptide treatment significantly increases recovery for GFP^{MYR5'/3'}nrrn1.

5) Most of the experiments are based on the overexpression of G3BP1 domains or treatment with Tat-fusion proteins while only axon length is quantified for the knockdown. The effects of a knockdown on translation and SGs should also be tested.

We have now included puromycinylation data for axonal protein synthesis in the G3BP1 knockdown neurons. siRNA depletion of G3BP1 decreases intra-axonal proteins synthesis (**Fig. 5c and Ext. Data Fig. 7a**).

Minor points

1) The references (6, 7) for "The RasGAP SH3 domain binding protein 1 (G3BP1) interacts with the 48S pre-initiation complex when translation is stalled, and it assembles SGs by virtue of its NTF2-like domain" (p. 3) appear to be wrong.

We have fixed these references. Thank you for catching our oversight.

2) The results in Ext. Data should be arranged in the same order as they are mentioned in the results (e.g. p. 5, Ext. Data Fig. S5B).

We have rearranged the extended data files to follow the results section as closely as possible. This of course increased the number of extended data files.

3) Fig. 5f: the cultures are incubated with the Tat-fusion peptide for 15 min. It is surprising that this is sufficient to have an effect. As a control no peptide is added. Instead an inactive peptide has to be used as control.

We were surprised by the rapid effect as well. We have now have included data with the control peptide in **Fig. 6d-f**. Notably, this effect represents disassembly of already assembled stress granule-like aggregates, while effects on protein translation and axon growth would be obviously take much longer.

REVIEWERS' COMMENTS:

Reviewer #1 (Remarks to the Author):

The revised manuscript has addressed most of my original points of criticism. Few minor points remain

1. I can follow the arguments why lesion experiments in G3BP1 deficient mice might not be suitable to show its role after nerve lesion. However, the previous studies by Zekri et al 2005 and Martin et al 2013 should be discussed and cited. Neglecting them in the context of this study is inadequate.
2. The new data in Fig.1e and 1f show that only a minor fraction of G3BP1 immunoreactive structures colabel with TIA1 . What is the role of the major fraction of G3BP1 which is not associated with TIO1 labeled stress granules? This needs to be convincingly explained.
3. I consider the reviewer figure 2 are very informative and recommend the authors to include this information in the suppl. file.

Reviewer #2 (Remarks to the Author):

The authors have made very comprehensive and appropriate responses to the comments of the reviewers. I have no further comments.

This is an excellent paper which is a major step forward in an important topic.

Reviewer #3 (Remarks to the Author):

The revised manuscript addresses many of the questions raised by the reviewers. However, a few points remain to be corrected.

- 1) The results shown in Fig. 1 do not convincingly demonstrate that G3BP1 is a component of SGs in axons. The authors calculated Pearson's coefficient to assess the co-localization of G3BP1 with different markers. They now include a statistical analysis to show that the difference in the values for the co-localization of G3BP1 with SG and PB markers is significant. Even if the difference is significant this does not mean it is biological meaningful. The values are rather low (around 0.2), which indicates a weak to moderate correlation and suggests that only a small fraction of G3BP1 co-localizes with SG markers. G3BP1 also shows very little co-localization with the SG marker TIA1 (Fig. 1e). It is more accurate to describe these structures as G3BP1 aggregates. The authors have to provide more convincing results to claim that G3BP1 forms stress granules in axons or modify their manuscript to clearly indicate that only a small fraction of G3BP1 localizes to SGs.
- 2) The controls to demonstrate the specificity of anti-G3BP1 antibodies in immunofluorescence staining are mentioned only as data not shown. They have to be included in the supplemental data.
- 3) References 6-8 were not replaced and are still incorrect.

NCOMMS-17-33728 - Revision # 2 – Response to Reviewers

We appreciate the reviewers' careful assessment of our revised manuscript and are delighted with a positive response. We outline the changes in the sections below with regards to the individual reviewers' comments (in red font).

Reviewer #1 (Remarks to the Author):

1. I can follow the arguments why lesion experiments in G3BP1 deficient mice might not be suitable to show its role after nerve lesion. However, the previous studies by Zekri et al 2005 and Martin et al 2013 should be discussed and cited. Neglecting them in the context of this study is inadequate.

We now reference the work of Zekri et al. and Martin et al. in the introduction (page 3).

2. The new data in Fig.1e and 1f show that only a minor fraction of G3BP1 immunoreactive structures colabel with TIA1. What is the role of the major fraction of G3BP1 which is not associated with TIO1 labeled stress granules? This needs to be convincingly explained.

We have added the following text to discuss possible roles for non-TIA1 associated G3BP1 on page 13 of the revised manuscript.

'Our study notably shows that axonal G3BP1 and TIA1 do not always colocalize. Likewise, the Pearson's coefficient for colocalization of G3BP1 with SG components is low despite being statistical higher than the coefficients for G3BP1 with PB proteins. Overexpression of G3BP1 was shown to precipitate SG assembly in the absence of any stress in non-neuronal cells¹. Only some of the aggregates seen with overexpressed G3BP1 colocalize with TIA1, but G3BP1-associating mRNAs were found in both TIA1-positive and TIA1-negative G3BP1 aggregates². Thus, it is likely that the axonal G3BP1 aggregates seen here that are separate from TIA1 can also interact with mRNAs. Regardless of whether the axonal G3BP1 aggregates are classic SGs or even core SG aggregates, our data clearly show these axonal aggregates attenuate axonal protein synthesis and limit rates of axon growth, so the axonal G3BP1 aggregates are biologically significant. Future studies will be needed to compare and contrast the constituents of these axonal G3BP1 aggregates to those of classic SGs.'

3. I consider the reviewer figure 2 are very informative and recommend the authors to include this information in the suppl. file.

We now include the figure as Supplementary Figure 5c.

Reviewer #2 (Remarks to the Author):

*The authors have made very comprehensive and appropriate responses to the comments of the reviewers. I have no further comments.
This is an excellent paper which is a major step forward in an important topic.*

Nothing to change here. Thanks!

Reviewer #3 (Remarks to the Author):

1) The results shown in Fig. 1 do not convincingly demonstrate that G3BP1 is a component of SGs in axons. The authors calculated Pearson's coefficient to assess the co-localization of G3BP1 with different markers. They now include a statistical analysis to show that the difference in the values for the co-localization of G3BP1 with SG and PB markers is significant. Even if the difference is significant this does not mean it is biological meaningful. The values are rather low (around 0.2), which indicates a weak to moderate correlation and suggests that only a small fraction of G3BP1 co-localizes with SG markers. G3BP1 also shows very little co-localization with the SG marker TIA1 (Fig. 1e). It is more accurate to describe these structures as G3BP1 aggregates. The authors have to provide more convincing results to claim that G3BP1 forms stress granules in axons or modify their manuscript to clearly indicate that only a small fraction of G3BP1 localizes to SGs.

As noted for Reviewer 1, point # 2 (see above) and suggested by the Editor, we have added text to address these issues on page 13. We also reference work on core stress granule components that have recently been uncovered by mass spec and emphasize that more work is needed on the axonal G3BP1 aggregates. Finally, we have tried to avoid the term stress granule throughout the manuscript, referring to these as SG-like or G3BP1 aggregates.

2) The controls to demonstrate the specificity of anti-G3BP1 antibodies in immunofluorescence staining are mentioned only as data not shown. They have to be included in the supplemental data.

We now include these in the Supplementary Figure 1c.

3) References 6-8 were not replaced and are still incorrect.

We confused the reviewer with separate reference lists for the main text and methods. A single reference list is now provided. Those references are now correct (now references 6, 7, & 10).